# Protein-Ligand Interaction Prior for Binding-aware 3D Molecule Diffusion Models

**Zhilin Huang**[1,2*]  **Ling Yang**[3*]  **Xiangxin Zhou**[4]  **Zhilong Zhang**[3]  **Wentao Zhang**[3]
**Xiawu Zheng**[2,5]  **Jie Chen**[2,3]  **Yu Wang**[2†]  **Cui Bin**[3]  **Wenming Yang**[1,2†]
[1] Shenzhen International Graduate School, Tsinghua University [2] Peng Cheng Laboratory
[3] Peking University [4] University of Chinese Academy of Sciences [5] Xiamen University
`{zerinhwang03,zzl2018math,wentao.zhang,bin.cui}@pku.edu.cn,`
`{yangling0818,yangelwm}@163.com, zhouxiangxin1998@gmail.com`
`zhengxiawu@xmu.edu.cn, {chenj,wangy20}@pcl.ac.cn`

## Abstract

Generating 3D ligand molecules that bind to specific protein targets via diffusion models has shown great promise for structure-based drug design. The key idea is to disrupt molecules into noise through a fixed forward process and learn its reverse process to generate molecules from noise in a denoising way. However, existing diffusion models primarily focus on incorporating protein-ligand interaction information solely in the reverse process, and neglect the interactions in the forward process. The inconsistency between forward and reverse processes may impair the binding affinity of generated molecules towards target protein. In this paper, we propose a novel **I**nteraction **P**rior-guided **Diff**usion model (IPDIFF) for the protein-specific 3D molecular generation by introducing geometric protein-ligand interactions into both diffusion and sampling process. Specifically, we begin by pretraining a protein-ligand interaction prior network (IPNET) by utilizing the binding affinity signals as supervision. Subsequently, we leverage the pretrained prior network to (1) integrate interactions between the target protein and the molecular ligand into the forward process for adapting the molecule diffusion trajectories (**prior-shifting**), and (2) enhance the binding-aware molecule sampling process (**prior-conditioning**). Empirical studies on CrossDocked2020 dataset show IPDIFF can generate molecules with more realistic 3D structures and state-of-the-art binding affinities towards the protein targets, with up to **-6.42** Avg. Vina Score, while maintaining proper molecular properties. https://github.com/YangLing0818/IPDiff

## 1 Introduction

Structure-based drug design (SBDD) (Anderson, 2003) plays an important role in drug discovery. Given a target protein and its 3D structure, we need to design ligand molecules *in silico* with desired properties, such as high binding affinity to the target. This problem can be formulated as a conditional generation task. Powerful deep generative models have already achieved promising results in SBDD tasks. For example, Luo et al. (2021); Liu et al. (2022); Peng et al. (2022) propose to generate atoms (and bonds) in an autoregressive way to form a ligand molecule, and Zhang et al. (2023) propose to generate 3D molecules fragment by fragment. Particularly, recent diffusion models (Sohl-Dickstein et al., 2015; Ho et al., 2020) exhibit remarkable abilities to synthesize realistic ligand molecules with high binding affinity to target proteins (Guan et al., 2023a; Lin et al., 2022; Schneuing et al., 2022; Guan et al., 2023b). They usually perturb atom types and positions of ligand molecules (**in the forward process**), and train an SE(3)-equivariant neural network to denoise the atom positions (resp. types) considering the interactions with target protein (**in the reverse process**) for fitting the reversal of the forward process.

We find that there is a discrepancy between the forward process and the reverse process regarding the utilization of interactions between the target protein and the generated molecular ligands, which

---

*Equal Contribution
†Corresponding Author

may limit the performance of the diffusion models for SBDD tasks. In the forward process, the ways of injecting noises are the same for all training samples with different target proteins. In other words, the differences of pocket binding sites between different training samples are neglected and all the molecules are perturbed in the same way during the forward process. However, in the reverse process, to generate ligand molecules that bind to specific receptors, the differences in pocket binding sites are considered. Such a discrepancy introduces a bias that hinders the diffusion models from fully capturing the interaction between pockets and ligand molecules, while such intermolecular interaction is the essence of pocket-ligand binding.

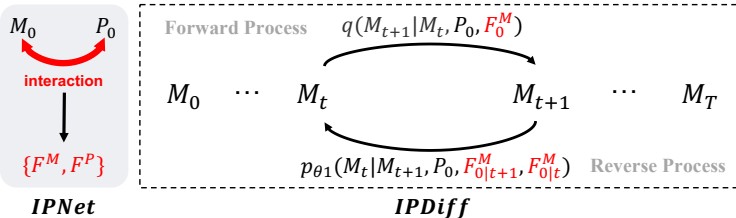

Figure 1: An illustration of forward (Equation (8)) and reverse diffusion steps (Equation (11)) incorporated with the interaction prior in IPDIFF, where $M_t$ indicates the molecular ligand at time step $t$, $P_0$ indicates given protein pocket and $F$ indicates the modeled interaction prior modeled by IPNET as described in Sec. 4.1. The difference between previous diffusion-based SBDD methods and IPDIFF are highlighted in red.

To eliminate this discrepancy, we propose a novel **I**nteraction **P**rior-guided **Diff**usion model (IPDIFF) to adaptively adjust the diffusion trajectories based on the specification of pocket-ligand interaction prior as illustrated in Figure 1. Specifically, the pocket-ligand interaction prior is captured with a pretrained neural network, named IPNET, which is supervised by binding affinity signals. Then we design an effective learnable adapter to explicitly incorporate such pocket-ligand interactions into all the timesteps in both forward and reverse processes for binding-aware trajectory adaptation (**prior-shifting**). Thus the two processes are jointly optimized during training, and IPDIFF is theoretically able to achieve better likelihood compared to previous molecular diffusion models. Besides, we propose to enhance the denoising by explicit conditioning on the estimated protein-ligand complex at the previous step in the reverse process (**prior-conditioning**), which is also powered by our learned interaction prior network. To demonstrate the efficacy of our IPDIFF, we conduct extensive experiments on CrossDocked2020 dataset. Empirical results show that our IPDIFF can generate ligands that not only bind tightly to target pockets but also maintain proper molecular properties, outperforming existing diffusion-based molecular diffusion models.

We highlight our main contributions as follows:

- We propose a novel 3D molecular diffusion model (IPDIFF) for SBDD where the pocket-ligand interaction is explicitly considered in both forward and reverse processes.
- We propose **prior-shifting** that shifts the diffusion trajectories of the forward process based on the interactions between pocket-binding sites and corresponding ligand molecules.
- We design **prior-conditioning** to enhance the reverse process by conditioning the denoising of ligand molecules on the previously estimated protein-ligand interactions.
- IPDIFF achieves SOTA performance on CrossDocked2020 benchmark, and it can generate the molecules with **-6.42 Avg. Vina Score** while maintaining proper molecular properties.

## 2 RELATED WORK

**Structure-based Drug Design**   Structure-based drug design (Anderson, 2003) aims to generate ligand molecules that bind to a given target protein and plays a critical role in the process of drug discovery. Skalic et al. (2019); Xu et al. (2021) proposed to generate ligand molecules in the format of SMILES conditioned on protein contexts. Tan et al. (2022) designed a flow model that can generate validated molecular drugs in the format of 2D graph conditioned on sequence embedding of specific targets. Ragoza et al. (2022b) voxelized molecules in atomic density grids and utilized VAE to generate 3D ligand molecules on receptor binding sites. Luo et al. (2021); Liu et al. (2022); Peng et al. (2022) proposed to generate atoms (and bonds) in 3D Euclidean space in an autoregressive

way. Zhang et al. (2023) proposed to generate 3D ligand molecules fragment by fragment. Recently, diffusion models have been applied to SBDD and achieved promising performance, which will be introduced in more detail in the following paragraph. Our work focuses on improving 3D molecular diffusion models for SBDD.

**Diffusion Models for SBDD**   Guan et al. (2023a); Lin et al. (2022); Schneuing et al. (2022) employ diffusion models to first generate atom types and positions, and then define the bonds as post-processing. They propose to utilize SE(3)-equivariant neural network to denoise the ligand molecules in the context of the protein-ligand complex, where the protein pocket is fixed. Schneuing et al. (2022) additionally tries to generate compounds (i.e. the protein-ligand complex) by inpainting conditioned on the pocket binding site. Guan et al. (2023b) further introduces decomposed priors into diffusion models for SBDD inspired by traditional drug discovery and achieves high binding affinity on average. Binding affinity, a critical evaluation metric in the process of drug discovery, measures whether drugs bind their target proteins selectively and specifically. Though DecompDiff (Guan et al., 2023b) has achieved exciting results, it heavily relies on external computation tools, which employs AlphaSpace2 (Rooklin et al., 2015) to extract subpockets and produce pocket priors when generating ligand molecules for new pockets. Differently from all the above, our molecular diffusion model for the first time considers pockets, ligand molecules, and their interactions in both forward and reverse processes. Moreover, our IPDIFF has no dependency on any external tools and achieves higher binding affinities than all other methods.

## 3   PRELIMINARY

The SBDD task from the perspective of generative models can be defined as generating ligand molecules which can bind to a given protein binding site. The target (protein) and ligand molecule can be represented as $\mathcal{P} = \{(\boldsymbol{x}_i^{\mathcal{P}}, \boldsymbol{v}_i^{\mathcal{P}})\}_{i=1}^{N_P}$ and $\mathcal{M} = \{(\boldsymbol{x}_i^{\mathcal{M}}, \boldsymbol{v}_i^{\mathcal{M}})\}_{i=1}^{N_M}$, respectively. Here $N_P$ (resp. $N_M$) refers to the number of atoms of the protein $\mathcal{P}$ (resp. the ligand molecule $\mathcal{M}$). $\boldsymbol{x} \in \mathbb{R}^3$ and $\boldsymbol{v} \in \mathbb{R}^K$ denote the position and type of the atom respectively. In the sequel, matrices are denoted by uppercase boldface. For a matrix $\mathbf{X}$, $\mathbf{x}_i$ denotes the vector on its $i$-th row, and $\mathbf{X}_{1:N}$ denotes the submatrix comprising its 1-st to $N$-th rows. For brevity, the ligand molecule is denoted as $\mathbf{M} = [\mathbf{X}^{\mathcal{M}}, \mathbf{V}^{\mathcal{M}}]$ where $\mathbf{X}^{\mathcal{M}} \in \mathbb{R}^{N_M \times 3}$ and $\mathbf{V}^{\mathcal{M}} \in \mathbb{R}^{N_M \times K}$, and the protein is denoted as $\mathbf{P} = [\mathbf{X}^{\mathcal{P}}, \mathbf{V}^{\mathcal{P}}]$ where $\mathbf{X}^{\mathcal{P}} \in \mathbb{R}^{N_P \times 3}$ and $\mathbf{V}^{\mathcal{P}} \in \mathbb{R}^{N_P \times K}$. The task can be formulated as modeling the conditional distribution $p(\mathbf{M}|\mathbf{P})$. Recently, diffusion models (Ho et al., 2020; Rombach et al., 2022; Song et al., 2020) have achieved promising performance in SBDD tasks (Guan et al., 2023a; Schneuing et al., 2022; Lin et al., 2022). The types and positions of the ligand molecular atoms are modeled by DDPMs (Ho et al., 2020), while the number of atoms $N_M$ is usually sampled from an empirical distribution (Hoogeboom et al., 2022; Guan et al., 2023a) or predicted by a neural network (Lin et al., 2022), and the chemical bonds are generated by the post-processing programs. And we define $\beta_t$ ($t = 1, \dots, T$) as fixed variance schedules, $\alpha_t = 1 - \beta_t$, $\bar{\alpha}_t = \prod_{s=1}^{t} \alpha_s$, $\bar{\beta}_t = 1 - \bar{\alpha}_t$. More detailed molecular diffusion processes can be found in Appendix A.

## 4   METHODS

As discussed in previous sections, we aim to incorporate protein-ligand interaction prior into 3D molecular diffusion model for generating ligand molecules binding tightly to the given pockets. We therefore propose IPDIFF, a novel diffusion-based model for binding-aware 3D molecule generation. We first design a prior network IPNET to capture the interactions between pockets and ligands from the perspective of both 3d structures and chemical properties, and pretrain it by binding affinity signals (Sec. 4.1). Then, we take the pretrained IPNET as interaction prior to facilitate the binding-aware ligand diffusion process. Two mechanisms, **prior-conditioning** and **prior-shifting** (Sec. 4.2), are proposed in IPDIFF to fully utilize the protein-molecule interactions in both forward and reverse processes of our diffusion framework.

### 4.1   LEARNING PROTEIN-LIGAND INTERACTION PRIOR WITH IPNET

IPNET consists of SE(3)-equivariant neural networks (Satorras et al., 2021) and cross-attention layers (Borgeaud et al., 2022; Hou et al., 2019). Two shallow fully-connected SE(3)-equivariant neural

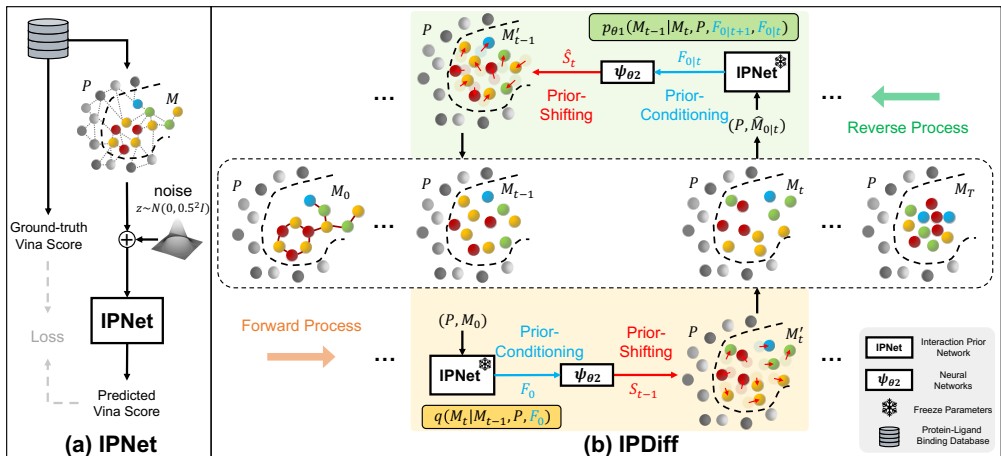

Figure 2: The overall schematic diagram of IPDIFF. The pretrained IPNET are frozen during both training and sampling process for providing interaction priors. The molecule $\mathbf{M}_0$ and $\hat{\mathbf{M}}_{0|t}$ are utilized for extracting interaction prior in forward and reverse process, respectively. The $\hat{\mathbf{M}}_{0|t}$ is the estimated molecule at the time step $t$ of the sampling process due to the inaccessibility of $\mathbf{M}_0$, $\mathbf{F}$ denotes the interactive representations and $\mathbf{S}$ denotes the position shifts of molecules.

networks are applied on the graphs of the protein $\mathcal{G}^{\mathcal{P}}$ and ligand molecule $\mathcal{G}^{\mathcal{M}}$, respectively, to learn the intramolecular interactions. And another shallow fully-connected SE(3)-equivariant neural network is applied on the graphs of the complexes graph $\mathcal{G}^{\mathcal{C}} = \mathcal{G}^{\mathcal{M}} \cup \mathcal{G}^{\mathcal{P}}$ which is constructed by $\mathcal{G}^{\mathcal{M}}$ and $\mathcal{G}^{\mathcal{P}}$, in order to modeling the intermolecular interactions. Given a ligand graph $\mathcal{G}^{\mathcal{M}}$, the $l$-th SE(3)-equivariant layer works as follows:

$$\mathbf{h}_i^{\mathcal{M},l+1} = \mathbf{h}_i^{\mathcal{M},l} + \sum_{j \in \mathcal{N}_M(i)} f_h^{\mathcal{M},l}\left(\left\|\mathbf{x}_i^{\mathcal{M},l} - \mathbf{x}_j^{\mathcal{M},l}\right\|, \mathbf{h}_i^{\mathcal{M},l}, \mathbf{h}_j^{\mathcal{M},l}\right), \tag{1}$$

$$\mathbf{x}_i^{\mathcal{M},l+1} = \mathbf{x}_i^{\mathcal{M},l} + \sum_{j \in \mathcal{N}_M(i)} \left(\mathbf{x}_i^{\mathcal{M},l} - \mathbf{x}_j^{\mathcal{M},l}\right) f_x^{\mathcal{M},l}\left(\left\|\mathbf{x}_i^{\mathcal{M},l} - \mathbf{x}_j^{\mathcal{M},l}\right\|, \mathbf{h}_i^{\mathcal{M},l+1}, \mathbf{h}_j^{\mathcal{M},l+1}\right), \tag{2}$$

where $\mathbf{h}_i^{\mathcal{M},l+1} \in \mathbb{R}^d$ and $\mathbf{x}_i^{\mathcal{M},l+1} \in \mathbb{R}^3$ are the SE(3)-invariant and SE(3)-equivariant hidden states of the atom $i$ of the ligand after the $l$-th SE(3)-equivariant layer, respectively. $\mathcal{N}_M(i)$ stands for the set of neighbors of atom $i$ on $\mathcal{G}^{\mathcal{M}}$, and the initial hidden state $\mathbf{h}_i^{\mathcal{M},0}$ is obtained by an embedding layer that encodes atom information. Given a protein graph $\mathcal{G}^{\mathcal{P}}$, $\mathbf{h}_i^{\mathcal{P},l}$, $\mathbf{x}_i^{\mathcal{P},l}$ and a complex graph $\mathcal{G}^{\mathcal{C}}$, $\mathbf{h}_i^{\mathcal{C},l}$, $\mathbf{x}_i^{\mathcal{C},l}$ can be derived in the same way.

In IPNET, an atom-wise cross-attention layer is introduced to model both intramolecular and intermolecular interactions of protein-ligand pairs, which essentially accounts for the binding affinity. The SE(3)-invariant features $\mathbf{H}^{\mathcal{M},L} \in \mathbb{R}^{N_M \times d}$ and $\mathbf{H}^{\mathcal{P},L} \in \mathbb{R}^{N_P \times d}$ are firstly concatenated along with the first dimension, and then concatenated with the SE(3)-invariant features $\mathbf{H}^{\mathcal{C},L} \in \mathbb{R}^{(N_P+N_M) \times d}$ along with channel dimension and finally fed into MLP:

$$[\![\tilde{\mathbf{F}}^{\mathcal{M}}, \tilde{\mathbf{F}}^{\mathcal{P}}]\!] = \mathrm{MLP}(\mathrm{Concat}([[\![\mathbf{H}^{\mathcal{M},L_1}, \mathbf{H}^{\mathcal{P},L_1}]\!], \mathbf{H}^{\mathcal{C},L_1}])) \tag{3}$$

where $[\![\cdot]\!]$ denotes concatenation along the first dimension. Then, $\mathbf{F}^{\mathcal{M}}$ and $\mathbf{F}^{\mathcal{P}}$ are delivered to the cross-attention layer for extracting interactions:

$$\mathbf{F}^{\mathcal{M}} = \mathrm{Attn}(\mathrm{Query}(\tilde{\mathbf{F}}^{\mathcal{M}}), \mathrm{Key}(\tilde{\mathbf{F}}^{\mathcal{P}})) \cdot \mathrm{Value}(\tilde{\mathbf{F}}^{\mathcal{P}}) \tag{4}$$

$$\mathbf{F}^{\mathcal{P}} = \mathrm{Attn}(\mathrm{Query}(\tilde{\mathbf{F}}^{\mathcal{P}}), \mathrm{Key}(\tilde{\mathbf{F}}^{\mathcal{M}})) \cdot \mathrm{Value}(\tilde{\mathbf{F}}^{\mathcal{M}}) \tag{5}$$

where the functions $\mathrm{Attn}$, $\mathrm{Query}$, $\mathrm{Key}$, and $\mathrm{Value}$ compute the cross attention weights and the query, key, and value matrices, respectively. The interactive representations of the ligand molecule $\mathbf{F}^{\mathcal{M}}$ and protein $\mathbf{F}^{\mathcal{P}}$ are further aggregated into a global feature to predict their binding affinity: $S_{\mathrm{Aff}}(\mathcal{P}, \mathcal{M}) \coloneqq \mathrm{IPNET}(\mathcal{P}, \mathcal{M})$. Please refer to Appendix D.1 for pre-training details. Next, we will describe how to utilize the prior network IPNET to facilitate the 3D molecular diffusion generation with protein-molecule interactions in both the forward and reverse processes.

## 4.2 PRIOR-GUIDED 3D MOLECULAR DIFFUSION MODEL

### 4.2.1 PRIOR-SHIFTING

We propose *prior-shifting* to shift the molecular diffusion trajectories on the positions of the ligand molecules based on the protein-molecule interactions modeled by IPNET. Firstly, IPDIFF feeds the ligand molecule $\mathbf{M}$ and the given pocket $\mathbf{P}$ into the pretrained prior network IPNET to extract the interactive representations of protein $\mathbf{F}^{\mathcal{P}}$ and molecule $\mathbf{F}^{\mathcal{M}}$. Then, a learnable neural network $\psi_{\theta 2}(\cdot)$ is introduced for producing the interaction-based shift for adapting the positions of ligand molecules. In practice, $\psi_{\theta 2}(\cdot)$ is 1-layer MLP which maps the updated molecule features with time step $t$ to the cumulative mean shift $\mathbf{S}_t^{\mathcal{M}} \in \mathbb{R}^{N_M \times 3}$ of the positions of ligand molecule:

$$\mathbf{S}_t^{\mathcal{M}} = \eta \cdot k_t \cdot \psi_{\theta 2}(\mathbf{F}^{\mathcal{M}}, t) \tag{6}$$

where $[\![\cdot]\!]$ denotes concatenation along the first dimension, $k_t$ is shift-mode coefficient and $\eta$ is shift-scale coefficient. We set the $k_t$ to $\sqrt{\bar{\alpha}} \cdot (1 - \sqrt{\bar{\alpha}})$. Since the shifts $S_t^{\mathcal{M}}$ are the cumulative values, this setting can keep the start point and the end point of the diffusion trajectories unchanged. $\eta$ is introduced to control the shift scales of diffusion trajectories, and we set $\eta = 1$ in our experiments.

More concretely, in the forward process, we adapt the molecular diffusion trajectories by injecting the learned binding-related molecule representation as a mean shift at each time step (the differences from the original molecular diffusion in Appendix A are highlighted in blue):

$$q(\mathbf{M}_t|\mathbf{M}_0, \mathbf{P}, \mathbf{F}_0^{\mathcal{M}}) = \prod_{i=1}^{N_M} \mathcal{N}(\mathbf{x}_{0,i}^{\mathcal{M}}; \sqrt{1 - \bar{\beta}_t}\mathbf{x}_{0,i}^{\mathcal{M}} + \mathbf{s}_{t,i}^{\mathcal{M}}, \bar{\beta}_t\boldsymbol{I}) \cdot \mathcal{C}(\mathbf{v}_{0,i}^{\mathcal{M}}|(1 - \bar{\beta}_t)\mathbf{v}_{0,i}^{\mathcal{M}} + \bar{\beta}_t/K), \tag{7}$$

$$q(\mathbf{M}_t|\mathbf{M}_{t-1}, \mathbf{P}, \mathbf{F}_0^{\mathcal{M}}) = \prod_{i=1}^{N_M} \mathcal{N}(\mathbf{x}_{t,i}^{\mathcal{M}}; \sqrt{1 - \beta_t}\mathbf{x}_{t-1,i}^{\mathcal{M}} + \mathbf{s}_{t,i}^{\mathcal{M}} - \sqrt{1 - \beta_t}\mathbf{s}_{t-1,i}^{\mathcal{M}}, \beta_t\boldsymbol{I}) \cdot$$
$$\mathcal{C}(\mathbf{v}_{t,i}^{\mathcal{M}}|(1 - \beta_t)\mathbf{v}_{t-1,i}^{\mathcal{M}} + \beta_t/K), \tag{8}$$

where $\mathcal{N}$ and $\mathcal{C}$ stand for the Gaussian and categorical distribution, respectively. And $\mathbf{s}_{t,i}^{\mathcal{M}}$ denotes the vector on the $i$-th row of $\mathbf{S}_t^{\mathcal{M}}$. We shall prove these two definitions are consistent in Appendix B. It is worth noting that, to fully exploit the interaction prior without introducing misleading noises, we utilize the interactive representations $\mathbf{F}_0^{\mathcal{M}}$ in Equations (6) to (8) in a teacher-forcing fashion, where $\mathbf{F}_0^{\mathcal{M}}$ is obtained by feeding the ground-truth protein-ligand pair $[\![\mathbf{X}_0^{\mathcal{M}}, \mathbf{X}_0^{\mathcal{P}}]\!], [\![\mathbf{V}_0^{\mathcal{M}}, \mathbf{V}_0^{\mathcal{P}}]\!]$ into the pretrained IPNET:

$$[\![\mathbf{F}_0^{\mathcal{M}}, \mathbf{F}_0^{\mathcal{P}}]\!] = \text{IPNET}([\![\mathbf{X}_0^{\mathcal{M}}, \mathbf{X}_0^{\mathcal{P}}]\!], [\![\mathbf{V}_0^{\mathcal{M}}, \mathbf{V}_0^{\mathcal{P}}]\!]) \tag{9}$$

Therefore, the corresponding shifted posterior can be analytically derived as follows:

$$q(\mathbf{M}_{t-1}|\mathbf{M}_t, \mathbf{M}_0, \mathbf{P}, \mathbf{F}_0^{\mathcal{M}}) = \prod_{i=1}^{N_M} \mathcal{N}(\mathbf{x}_{t-1,i}^{\mathcal{M}}; \tilde{\boldsymbol{\mu}}(\mathbf{x}_{t,i}^{\mathcal{M}}, \mathbf{x}_{0,i}^{\mathcal{M}}, \mathbf{f}_{0,i}^{\mathcal{M}}), \tilde{\beta}_t\boldsymbol{I}) \cdot$$
$$\mathcal{C}(\mathbf{v}_{t-1,i}^{\mathcal{M}}|\tilde{\boldsymbol{c}}(\mathbf{v}_{t,i}^{\mathcal{M}}, \mathbf{v}_{0,i}^{\mathcal{M}})), \tag{10}$$

where $\tilde{\boldsymbol{\mu}}(\mathbf{x}_{t,i}^{\mathcal{M}}, \mathbf{x}_{0,i}^{\mathcal{M}}, \mathbf{f}_{0,i}^{\mathcal{M}}) = \frac{\sqrt{\bar{\alpha}_{t-1}}\beta_t}{1 - \bar{\alpha}_t}\mathbf{x}_{0,i}^{\mathcal{M}} + \frac{\sqrt{\alpha_t}(1 - \bar{\alpha}_{t-1})}{1 - \bar{\alpha}_t}(\mathbf{x}_{t,i}^{\mathcal{M}} - \mathbf{s}_{t,i}^{\mathcal{M}}) + \mathbf{s}_{t-1,i}^{\mathcal{M}}$, $\tilde{\beta}_t = \frac{1 - \bar{\alpha}_{t-1}}{1 - \bar{\alpha}_t}\beta_t$, $\alpha_t = 1 - \beta_t$, $\bar{\alpha}_t = \prod_{i=1}^t \alpha_i$, $\tilde{\boldsymbol{c}}(\mathbf{v}_{t,i}, \mathbf{v}_{0,i}) = \frac{\boldsymbol{c}^*}{\sum_{k=1}^K c_k^*}$, and $\boldsymbol{c}^*(\mathbf{v}_{t,i}, \mathbf{v}_{0,i}) = [\alpha_t\mathbf{v}_{t,i} + (1 - \alpha_t)/K] \odot [\bar{\alpha}_{t-1}\mathbf{v}_{0,i} + (1 - \bar{\alpha}_{t-1})/K]$.

In the reverse process, since the ground-truth molecule $\mathbf{X}_0^{\mathcal{M}}$ and $\mathbf{V}_0^{\mathcal{M}}$ are inaccessible at the time step $t$, we utilize the molecule $\hat{\mathbf{M}}_{0|t+1} = [\hat{\mathbf{X}}_{0|t+1}^{\mathcal{M}}, \hat{\mathbf{V}}_{0|t+1}^{\mathcal{M}}]$ estimated in the previous time step $t + 1$ to substitute the $\mathbf{M}_0$ and feed it with $\mathbf{P}$ into the pretrained IPNET for obtaining the interactive representation $\mathbf{F}_{0|t+1}^{\mathcal{M}}$. And then we calculate the $\hat{\mathbf{S}}_t$ from $\mathbf{F}_{0|t+1}^{\mathcal{M}}$ according to the Equation (6). Similarly, we calculate the $\hat{\mathbf{S}}_{t-1}$ from $\mathbf{F}_{0|t}^{\mathcal{M}}$. Therefore, the reverse transition kernel can be approximated with predicted atom types $\hat{\mathbf{v}}_{0|t,i}$ and atom positions $\hat{\mathbf{x}}_{0|t,i}$ as follows:

$$p_{\theta 1}(\mathbf{M}_{t-1}|\mathbf{M}_t, \mathbf{P}, \mathbf{F}_{0|t+1}^{\mathcal{M}}, \mathbf{F}_{0|t}^{\mathcal{M}}) = \prod_{i=1}^{N_M} \mathcal{N}(\mathbf{x}_{t-1,i}^{\mathcal{M}}; \tilde{\boldsymbol{\mu}}(\mathbf{x}_{t,i}^{\mathcal{M}}, \hat{\mathbf{x}}_{0|t,i}^{\mathcal{M}}, \mathbf{f}_{0|t+1,i}^{\mathcal{M}}, \mathbf{f}_{0|t,i}^{\mathcal{M}}), \tilde{\beta}_t\boldsymbol{I}) \cdot$$
$$\mathcal{C}(\mathbf{v}_{t-1,i}^{\mathcal{M}}|\tilde{\boldsymbol{c}}(\mathbf{v}_{t,i}^{\mathcal{M}}, \hat{\mathbf{v}}_{0|t,i}^{\mathcal{M}})). \tag{11}$$

where $\tilde{\boldsymbol{\mu}}(\mathbf{x}_{t,i}^{\mathcal{M}}, \mathbf{x}_{0,i}^{\mathcal{M}}, \mathbf{f}_{0|t+1,i}^{\mathcal{M}}, \mathbf{f}_{0|t,i}^{\mathcal{M}}) = \frac{\sqrt{\bar{\alpha}_{t-1}}\beta_t}{1-\bar{\alpha}_t}\mathbf{x}_{0,i}^{\mathcal{M}} + \frac{\sqrt{\bar{\alpha}_t}(1-\bar{\alpha}_{t-1})}{1-\bar{\alpha}_t}(\mathbf{x}_{t,i}^{\mathcal{M}} - \hat{\mathbf{s}}_{t,i}^{\mathcal{M}}) + \hat{\mathbf{s}}_{t-1,i}^{\mathcal{M}}$. In this way, our IPDIFF can align molecular diffusion process with molecular sampling process regarding the information utilization of target protein, and optimize the diffusion trajectories according to the protein-molecule interaction.

### 4.2.2 PRIOR-CONDITIONING

In order to maximize the exploitation of protein-ligand interaction prior in the pretrained IPNET, we propose *prior-conditioning* to condition the molecular sampling process on previously estimated protein-molecule complex for facilitating the binding-aware molecular generation. Specifically, we leverage the ligand atom embedding $\mathbf{F}_{0|t+1}^{\mathcal{M}}$ and protein atom embedding $\mathbf{F}_{0|t+1}^{\mathcal{P}}$ extracted by the IPNET$(\hat{\mathbf{M}}_{0|t+1}, \mathbf{P})$ to enhance the embeddings at the time step $t$:

$$\mathbf{H}_t^{\mathcal{M},0} = \text{Concat}(\tilde{\mathbf{H}}_t^{\mathcal{M}}, \mathbf{F}_{0|t+1}^{\mathcal{M}}), \ \mathbf{H}_t^{\mathcal{P},0} = \text{Concat}(\tilde{\mathbf{H}}_t^{\mathcal{P}}, \mathbf{F}_{0|t+1}^{\mathcal{P}}). \tag{12}$$

where $\tilde{\mathbf{H}}_t^{\mathcal{M}}$ ($\tilde{\mathbf{H}}_t^{\mathcal{P}}$) denotes the initial hidden states of the ligand (protein) at the time stpe $t$ to be fed into the first SE(3)-equivariant layer of the diffusion model. Actually, the molecule $\hat{\mathbf{M}}_{0|t+1}$ estimated at time step $t+1$ is supposed to be a candidate ligand with high binding affinity towards target protein, especially when $t+1$ is large (*i.e.*, the generative process nearly ends). Due to the inaccessibility of $\hat{\mathbf{M}}_{0|t+1}$ during the training phase, we directly utilize the ground truth molecule $\mathcal{M}$ to substitute it in a teacher-forcing fashion.

**3D Equivariant Molecular Diffusion**  We then apply a neural network with $L_2$ layers on the $k$-nn graph of the protein-ligand complex enhanced by our prior-conditioning (denoted as $\mathbf{C} = [\![\mathbf{M}, \mathbf{P}]\!]$, where $[\![\cdot]\!]$ denotes concatenation along the first dimension) to learn the geometric interactions between the ligand atoms and the protein atoms. The hidden state $\mathbf{H}_t^{\mathcal{C}}$ and positions $\mathbf{X}_t^{\mathcal{C}}$ at the time step $t$ are updated as follows:

$$\mathbf{h}_{t,i}^{\mathcal{C},l+1} = \mathbf{h}_{t,i}^{\mathcal{C},l} + \sum_{j \in \mathcal{N}_C(i)} f_h^{\mathcal{C},l}\left(\left\|\mathbf{x}_{t,i}^{\mathcal{C},l} - \mathbf{x}_{t,j}^{\mathcal{C},l}\right\|, \mathbf{h}_{t,i}^{\mathcal{C},l}, \mathbf{h}_{t,j}^{\mathcal{C},l}, \mathbf{e}_{ij}^{\mathcal{C}}\right) \tag{13}$$

$$\mathbf{x}_{t,i}^{\mathcal{C},l+1} = \mathbf{x}_{t,i}^{\mathcal{C},l} + \sum_{j \in \mathcal{N}_C(i)} \left(\mathbf{x}_{t,i}^{\mathcal{C},l} - \mathbf{x}_{t,j}^{\mathcal{C},l}\right) f_x^{\mathcal{C},l}\left(\left\|\mathbf{x}_{t,i}^{\mathcal{C},l} - \mathbf{x}_{t,j}^{\mathcal{C},l}\right\|, \mathbf{h}_{t,i}^{\mathcal{C},l+1}, \mathbf{h}_{t,j}^{\mathcal{C},l+1}, \mathbf{e}_{ij}^{\mathcal{C}}\right) \cdot \mathbb{1}_{\text{mol}} \tag{14}$$

where $\mathcal{N}_C(i)$ stands for the set of $k$-nearest neighbors of atom $i$ on the protein-ligand complex graph, $\mathbf{e}_{ij}^{\mathcal{C}}$ indicates the atom $i$ and atom $j$ are both protein atoms or both ligand atoms or one protein atom and one ligand atom, and $\mathbb{1}_{\text{mol}}$ is the ligand atom mask since the protein atom coordinates are known and thus supposed to remain unchanged during this update. We let $\mathbf{H}_t^{\mathcal{C},0} := [\![\mathbf{H}_t^{\mathcal{M},0}, \mathbf{H}_t^{\mathcal{P},0}]\!]$ as the representations incorporate the interaction information through the prior-conditioning mechanism as described in Equation (12). Finally, we use $\hat{\mathbf{V}}_{0|1} = \text{softmax}(\text{MLP}(\mathbf{H}_{0,1:N_M}^{\mathcal{C},L_2}))$ and $\hat{\mathbf{X}}_{0|1} = \mathbf{X}_{0,1:N_M}^{\mathcal{C},L_2}$ as the final prediction. We leave the details about the loss function and summarize the training and sampling procedures of IPDIFF in Appendix C.

## 5 EXPERIMENTS

### 5.1 EXPERIMENTAL SETTINGS

**Datasets and Baseline Methods**  For fully modeling protein-ligand interactions, we utilize the protein-ligand pairs (complexes) in PDBbind v2016 dataset Liu et al. (2015) with binding affinity signals to pretrain the IPNET. The PDBbind v2016 dataset consists of 3767 training complexes and 290 testing complexes, and it is commonly employed in binding-affinity prediction tasks. For molecular generation, following the previous work Luo et al. (2021); Peng et al. (2022); Guan et al. (2023a), we train and evaluate IPDIFF on the CrossDocked2020 dataset (Francoeur et al., 2020). The same data preparation and splitting as Luo et al. (2021) are employed, where the 22.5 million docked binding complexes are refined to high-quality docking poses (RMSD between the docked pose and the ground truth $< 1$Å) and diverse proteins (sequence identity $< 30\%$). Specifically, $100,000$ protein-ligand pairs are utilized for training and $100$ proteins for testing. In our study, we conduct a comparative analysis of our model with five recent representative methods for SBDD. **LiGAN**

(Ragoza et al., 2022a) is a CVAE model trained on an atomic density grid representation of protein-ligand structures. **AR** (Luo et al., 2021) and **Pocket2Mol** (Peng et al., 2022) generate 3D molecules atoms conditioned on the protein pocket and previous generated atoms in an autoregressive manner. **TargetDiff** (Guan et al., 2023a) and **DecomposeDiff** (Guan et al., 2023b) are recent state-of-the-art diffusion methods which generate atom coordinates and atom types in a non-autoregressive way.

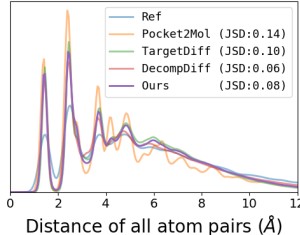

Figure 3: Comparing the distribution for distances of all-atom for reference molecules in the test set (blue) and model generated molecules (color). Jensen-Shannon divergence (JSD) between two distributions is reported.

| Bond | liGAN | AR | Pocket2 Mol | Target Diff | Decomp Diff | ours |
|------|-------|-----|-------------|-------------|-------------|------|
| C−C | 0.601 | 0.609 | 0.496 | 0.369 | **0.359** | 0.386 |
| C=C | 0.665 | 0.620 | 0.561 | 0.505 | 0.537 | **0.245** |
| C−N | 0.634 | 0.474 | 0.416 | 0.363 | 0.344 | **0.298** |
| C=N | 0.749 | 0.635 | 0.629 | 0.550 | 0.584 | **0.238** |
| C−O | 0.656 | 0.492 | 0.454 | 0.421 | 0.376 | **0.366** |
| C=O | 0.661 | 0.558 | 0.516 | 0.461 | 0.374 | **0.353** |
| C:C | 0.497 | 0.451 | 0.416 | 0.263 | 0.251 | **0.169** |
| C:N | 0.638 | 0.552 | 0.487 | 0.235 | 0.269 | **0.128** |

Table 1: Jensen-Shannon divergence between bond distance distributions of the reference molecules and the generated molecules, and lower values indicate better performances. "-", "=", and ":" represent single, double, and aromatic bonds, respectively.

**Evaluation**   Three perspectives: **molecular structures**, **target binding affinity** and **molecular properties** are considered for evaluating generated molecular ligands. The Jensen-Shannon divergences (JSD) in empirical distributions of atom/bond distances between the generated and the reference molecules are calculated for evaluating the generated molecules in terms of **molecular structures**. Following previous work Luo et al. (2021); Ragoza et al. (2022b); Guan et al. (2023a), AutoDock Vina (Eberhardt et al., 2021) is utilized to compute the mean and median of binding-related metrics, including *Vina Score*, *Vina Min*, *Vina Dock* and *High Affinity*. Vina Score directly estimates the binding affinity based on the generated 3D molecules; Vina Min performs a local structure minimization before estimation; Vina Dock involves an additional re-docking process and reflects the best possible binding affinity; High affinity measures the ratio of how many generated molecules binds better than the reference molecule per test protein. And we utilize the *QED*, *SA*, *Diversity* as metrics following Luo et al. (2021); Ragoza et al. (2022a) for evaluating **molecular properties**. QED is a simple quantitative estimation of drug-likeness combining several desirable molecular properties; SA (synthesize accessibility) is a measure estimation of the difficulty of synthesizing the ligands; Diversity is computed as average pairwise dissimilarity between all ligands generated by the given pocket. All sampling and evaluation procedures are following Guan et al. (2023a) for fair comparison.

## 5.2 MAIN RESULTS

**Generated Molecular Structures**   We compare the molecular structures of molecules generated by our IPDIFF and the other representative methods. The all-atom pairwise distance distribution of the generated molecules are plotted in Figure 3. And Tab. 1 presents the bond distributions of the molecules generated by different methods compared against the corresponding reference empirical distributions. And our IPDIFF achieves superior performance on major bond types compared to all other methods, which demonstrating the ability of IPDIFF in generating stable molecular structures.

**Target Binding Affinity and Molecule Properties**   We evaluate the effectiveness of IPDIFF by comparing with two types of SBDD methods: non-diffusion methods and diffusion-based methods in Tab. 2. Our IPDIFF significantly outperforms non-diffusion baselines in binding-related metrics. Notably, IPDIFF also surpasses strong autoregressive method Pocket2Mol by a large margin of **24.9%**, **16.0%** and **19.9%** in Avg. Vina Score, Vina Min and Vina Dock, respectively. Compared with the state-of-the-art diffusion-based method DecompDiff, IPDIFF not only increases the binding-related metrics Avg. Vina Score, Vina Min and Vina Dock by **13.2%**, **5.8%** and **2.1%**, but also increases the property-related metric Avg. QED and Avg. Diversity by **15.6%** and **8.8%**. In terms of high-affinity binder, we find that on average **69.5%** of the IPDIFF molecules show better binding affinity than the reference molecule, which is significantly better than other baselines. These gains demonstrate

that the proposed IPDIFF effectively utilize protein-ligand interaction priors from IPNET to enable generating molecules with improved target binding affinity and molecular property.

Table 2: Summary of different properties of reference molecules and molecules generated by our model and other non-diffusion (Non-Diff.) and diffusion-based (Diff.) baselines. (↑) / (↓) denotes a larger / smaller number is better. Top 2 results are highlighted with **bold text** and underlined text, respectively.

| Methods | | Vina Score (↓) | | Vina Min (↓) | | Vina Dock (↓) | | High Affinity (↑) | | QED (↑) | | SA (↑) | | Diversity (↑) | |
|---|---|---|---|---|---|---|---|---|---|---|---|---|---|---|---|
| | | Avg. | Med. | Avg. | Med. | Avg. | Med. | Avg. | Med. | Avg. | Med. | Avg. | Med. | Avg. | Med. |
| Reference | | -6.36 | -6.46 | -6.71 | -6.49 | -7.45 | -7.26 | - | - | 0.48 | 0.47 | 0.73 | 0.74 | - | - |
| Comp. with Non-Diff. | LiGAN | - | - | - | - | -6.33 | -6.20 | 21.1% | 11.1% | 0.39 | 0.39 | 0.59 | 0.57 | 0.66 | 0.67 |
| | GraphBP | - | - | - | - | -4.80 | -4.70 | 14.2% | 6.7% | 0.43 | 0.45 | 0.49 | 0.48 | **0.79** | **0.78** |
| | AR | -5.75 | -5.64 | -6.18 | -5.88 | -6.75 | -6.62 | 37.9% | 31.0% | 0.51 | 0.50 | 0.63 | 0.63 | 0.70 | 0.70 |
| | Pocket2Mol | -5.14 | -4.70 | -6.42 | -5.82 | -7.15 | -6.79 | 48.4% | 51.0% | **0.56** | **0.57** | **0.74** | **0.75** | 0.69 | 0.71 |
| | **IPDIFF** | **-6.42** | **-7.01** | **-7.45** | **-7.48** | **-8.57** | **-8.51** | **69.5%** | **75.5%** | 0.52 | 0.53 | 0.61 | 0.59 | 0.74 | 0.73 |
| Comp. with Diff. | TargetDiff | -5.47 | -6.30 | -6.64 | -6.83 | -7.80 | -7.91 | 58.1% | 59.1% | 0.48 | 0.48 | 0.58 | 0.58 | 0.72 | 0.71 |
| | DecompDiff | -5.67 | -6.04 | -7.04 | -7.09 | -8.39 | -8.43 | 64.4% | 71.0% | 0.45 | 0.43 | **0.61** | **0.60** | 0.68 | 0.68 |
| | **IPDIFF** | **-6.42** | **-7.01** | **-7.45** | **-7.48** | **-8.57** | **-8.51** | **69.5%** | **75.5%** | **0.52** | **0.53** | **0.61** | 0.59 | **0.74** | **0.73** |

**Achieving Better Trade-off** From Tab. 2, we can see a trade-off between binding-related metrics and property-related metrics QED in previous methods. DecompDiff performs better than AR and Pocket2Mol in binding-related metrics, but falls behind them in QED scores. In contrast, our IPDIFF not only achieves the state-of-the-art binding-related scores but also maintains proper QED score which is comparable to Pocket2Mol, achieving a better trade-off than DecompDiff. Nevertheless, we put less emphasis on QED and SA because they are often applied as rough screening metrics in real drug discovery scenarios, and it would be acceptable as long as they are within a reasonable range. Figure 4 shows some examples of generated ligand molecules and their properties. The molecules generated by our model have valid structures and reasonable binding poses to the target, which are supposed to be promising candidate ligands. **More ablation studies, experimental results and visualized examples of generated molecules are present in the Appendices E and G.**

## 5.3 MODEL ANALYSIS

**Effect of Pretrained IPNET** In IPDIFF, we leverage the protein-ligand interaction prior in pre-trained IPNET to facilitate the molecular diffusion. Here we conduct ablation study on the effect of our IPNET. We designed two types of IPNET: (1) IPNET, simultaneously considers the geometry and sequence information in modeling interaction priors by simply stacking SE(3)-Equivariant layer as described in Sec. 4.1, and (2) IPNET-Seq., replaces SE(3)-equivariant layers in IPNET with graph attention layers (Velickovic et al., 2017) and only considers the sequence information by taking $[\mathbf{V}^{\mathcal{M}}, \mathbf{V}^{\mathcal{P}}]$ as inputs, and then we applied these two types of IPNET into the training and sampling process of IPDIFF. The results are present in Tab. 3. We can observe that even if only the sequence-level interaction prior is utilized to guide the generation process of sequences and geometric structures of IPDIFF, IPDIFF can achieve superior performance than the baseline model, especially in the property-related metrics. It reveals that our IPDIFF does not rely heavily on the elaborate designs of the IPNET. Moreover, we jointly train the IPNET with IPDIFF from scratch without pre-training. We found that training whole model from scratch converges slowly on the same device that is used for all the experiments. And the final performance is consistently worse than that with pre-training. The main reason is that training generative models from scratch lacks explicit supervision (i.e., binding affinity) for IPNET to model the accurate interaction prior. This demonstrates that the modeled interaction prior through pretraining IPNET plays an important role in our method.

**Effectiveness of Prior-Conditioning and Prior-Shifting** Our primary hypothesis is that introducing the 3D protein-molecule interaction prior into both forward and reverse process benefits the training and sampling efficiency, and thus improving the molecular generation performance in both binding- and property-related metrics. To verify it, we conducting a set of experiments to showcase the effectiveness of prior-conditioning and prior-shifting. All results are present in Tab. 4. We observe that self-conditioning mechanism proposed by Chen et al. (2023) can not improve the generation performance, because the estimated molecules from previous time step does not include the protein-ligand interaction information for self refinement. In contrast, our prior-conditioning mechanism

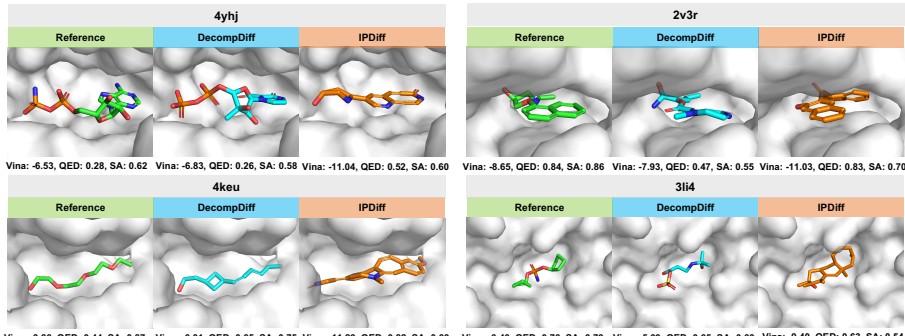

Figure 4: Examples of generated ligands for protein pockets (4yhj, 2v3r, 4keu and 3li4). Carbon atoms in reference ligands, ligands generated by DecompDiff (Guan et al., 2023b) and IPDIFF are visualized in green, blue and orange, respectively. Vina Score, QED and SA are reported.

significantly boosts both binding-related and property-related metrics by introducing informative protein-ligand interaction knowledge from our pretrained IPNET. Besides, our prior-shifting also has a notable improvement over baseline in binding-related metrics which reveals that prior-shifting can effectively assist IPDIFF to generate ligand molecules binding tightly to the given protein pockets. Kindly note that prior-shifting does significantly contribute to property-related metrics. This because the prior-shifting is only equipped in position of molecular atoms, while property-related metrics are less dependent to the geometry of the protein-ligand pair. Furthermore, simultaneously utilizing prior-conditioning and prior-shifting in IPDIFF yields the best performance in both binding-related and property-related metrics, which demonstrates the effectiveness of two mechanisms.

Table 3: Effect of the pretrained IPNET. (↑) / (↓) denotes a larger / smaller number is better. Top 2 results are highlighted with **bold text** and underlined text, respectively.

| Methods | Vina Score (↓) | | Vina Min (↓) | | Vina Dock (↓) | | High Affinity (↑) | | QED (↑) | | SA (↑) | |
|---|---|---|---|---|---|---|---|---|---|---|---|---|
| | Avg. | Med. | Avg. | Med. | Avg. | Med. | Avg. | Med. | Avg. | Med. | Avg. | Med. |
| baseline | -5.04 | -5.75 | -6.38 | -6.52 | -7.55 | -7.72 | 54.2% | 54.1% | 0.46 | 0.46 | 0.57 | 0.57 |
| IPNET w/o Pre-training | -3.88 | -4.98 | -5.38 | -5.80 | -6.96 | -7.20 | 43.2% | 34.0% | 0.39 | 0.39 | 0.58 | 0.57 |
| IPNET-Seq | -5.78 | -6.95 | -7.11 | **-7.53** | -8.34 | **-8.56** | 69.0% | **76.4%** | **0.57** | **0.58** | 0.55 | 0.54 |
| IPNET | **-6.42** | **-7.01** | **-7.45** | -7.48 | **-8.57** | -8.51 | **69.5**% | 75.5% | 0.52 | 0.53 | **0.61** | **0.59** |

Table 4: The effect of prior-conditioning and prior-shifting mechanism. (↑) / (↓) denotes a larger / smaller number is better. Top 2 results are highlighted with **bold text** and underlined text, respectively.

| Methods | Vina Score (↓) | | Vina Min (↓) | | Vina Dock (↓) | | High Affinity (↑) | | QED (↑) | | SA (↑) | |
|---|---|---|---|---|---|---|---|---|---|---|---|---|
| | Avg. | Med. | Avg. | Med. | Avg. | Med. | Avg. | Med. | Avg. | Med. | Avg. | Med. |
| baseline | -5.04 | -5.75 | -6.38 | -6.52 | -7.55 | -7.72 | 54.2% | 54.1% | 0.46 | 0.46 | 0.57 | 0.57 |
| baseline + self-conditioning | -4.94 | -6.06 | -6.18 | -6.46 | -7.43 | -7.55 | 53.3% | 51.7% | 0.46 | 0.49 | 0.56 | 0.56 |
| baseline + prior-conditioning | -5.30 | -6.17 | -6.61 | -6.81 | -7.94 | -8.01 | 60.6% | 66.9% | **0.52** | **0.53** | **0.62** | **0.61** |
| baseline + prior-shifting | -5.51 | -6.39 | -6.87 | -7.09 | -8.06 | -8.24 | 64.4% | 64.0% | 0.48 | 0.48 | 0.56 | 0.56 |
| IPDIFF | **-6.42** | **-7.01** | **-7.45** | **-7.48** | **-8.57** | **-8.51** | **69.5**% | **75.5**% | 0.52 | 0.53 | 0.61 | 0.59 |

## 6 CONCLUSION

In this paper, we for the first time introduce target protein into both diffusion and sampling process, and propose a novel **I**nteraction **P**rior-guided **Diff**usion model (IPDIFF) for the protein-specific 3D molecular generation. We design IPNET to learn protein-ligand interaction prior with the supervision of binding affinity signals, which further is utilized to facilitate the binding-aware 3D molecular diffusion generation with our proposed *prior-shifting* and *prior-conditioning*. Empirical studies on CrossDocked2020 dataset show IPDIFF can generate molecules with more realistic 3D structures and state-of-the-art binding affinities towards the protein targets, with up to **-6.42** Avg. Vina Score, while maintaining proper molecular properties. Moreover, we conduct extensive analysis experiments to demonstrate the effectiveness and superiority of the proposed model.

## ACKNOWLEDGMENTS

The work was partly supported by the National Natural Science Foundation of China (No.62171251), the Major Key Project of GZL under Grant SRPG22-001, the Major Key Research Project of PCL under Grant PCL2023A08 and PCL2023A09.

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

## A  CONVENTIONAL PROTEIN-AWARE 3D MOLECULAR DIFFUSION PROCESS

In the forward diffusion process, a small Gaussian noise is gradually injected into data as a Markov chain. Because noises are only added on ligand molecules but not proteins in the diffusion process, we denote the atom positions and types of the ligand molecule at time step $t$ as $\mathbf{X}_t^{\mathcal{M}}$ and $\mathbf{V}_t^{\mathcal{M}}$. The diffusion transition kernel can be defined as follows:

$$q(\mathbf{M}_t|\mathbf{M}_{t-1},\mathbf{P}) = \prod_{i=1}^{N_M} \mathcal{N}(\mathbf{x}_{t,i}^{\mathcal{M}}; \sqrt{1-\beta_t}\mathbf{x}_{t-1,i}^{\mathcal{M}}, \beta_t\mathbf{I}) \cdot \mathcal{C}(\mathbf{v}_{t,i}^{\mathcal{M}}|(1-\beta_t)\mathbf{v}_{t-1,i}^{\mathcal{M}} + \beta_t/K), \quad (15)$$

$$q(\mathbf{M}_t|\mathbf{M}_0,\mathbf{P}) = \prod_{i=1}^{N_M} \mathcal{N}(\mathbf{x}_{i,0}^{\mathcal{M}}; \sqrt{1-\bar{\beta}_t}\mathbf{x}_{i,0}^{\mathcal{M}}, \bar{\beta}_t\mathbf{I}) \cdot \mathcal{C}(\mathbf{v}_{i,0}^{\mathcal{M}}|(1-\bar{\beta}_t)\mathbf{v}_{i,0}^{\mathcal{M}} + \bar{\beta}_t/K), \quad (16)$$

where $\mathcal{N}$ and $\mathcal{C}$ stand for the Gaussian and categorical distribution respectively, $\beta_t$ is defined by fixed variance schedules. The corresponding posterior can be analytically derived as follows:

$$q(\mathbf{M}_{t-1}|\mathbf{M}_t,\mathbf{M}_0,\mathbf{P}) = \prod_{i=1}^{N_M} \mathcal{N}(\mathbf{x}_{t-1,i}^{\mathcal{M}}; \tilde{\boldsymbol{\mu}}(\mathbf{x}_{t,i}^{\mathcal{M}},\mathbf{x}_{0,i}^{\mathcal{M}}), \tilde{\beta}_t\mathbf{I}) \cdot \mathcal{C}(\mathbf{v}_{t-1,i}^{\mathcal{M}}|\tilde{\boldsymbol{c}}(\mathbf{v}_{t,i}^{\mathcal{M}},\mathbf{v}_{0,i}^{\mathcal{M}})), \quad (17)$$

where $\tilde{\boldsymbol{\mu}}(\mathbf{x}_{t,i}^{\mathcal{M}},\mathbf{x}_{0,i}^{\mathcal{M}}) = \frac{\sqrt{\bar{\alpha}_{t-1}}\beta_t}{1-\bar{\alpha}_t}\mathbf{x}_{0,i}^{\mathcal{M}} + \frac{\sqrt{\alpha_t}(1-\bar{\alpha}_{t-1})}{1-\bar{\alpha}_t}\mathbf{x}_{t,i}^{\mathcal{M}}$, $\tilde{\beta}_t = \frac{1-\bar{\alpha}_{t-1}}{1-\bar{\alpha}_t}\beta_t$, $\alpha_t = 1-\beta_t$, $\bar{\alpha}_t = \prod_{s=1}^{t}\alpha_s$, $\tilde{\boldsymbol{c}}(\mathbf{v}_{t,i}^{\mathcal{M}},\mathbf{v}_{0,i}^{\mathcal{M}}) = \frac{\boldsymbol{c}^*}{\sum_{k=1}^{K} c_k^*}$, and $\boldsymbol{c}^*(\mathbf{v}_{t,i}^{\mathcal{M}},\mathbf{v}_{0,i}^{\mathcal{M}}) = [\alpha_t\mathbf{v}_{t,i}^{\mathcal{M}} + (1-\alpha_t)/K] \odot [\bar{\alpha}_{t-1}\mathbf{v}_{0,i}^{\mathcal{M}} + (1-\bar{\alpha}_{t-1})/K]$.

In the approximated reverse process, also known as the generative process, a neural network parameterized by $\theta 1$ learns to recover data by iteratively denoising. The reverse transition kernel can be approximated with predicted atom types $\hat{\mathbf{v}}_{0|t,i}^{\mathcal{M}}$ and atom positions $\hat{\mathbf{x}}_{0|t,i}^{\mathcal{M}}$ at time step $t$ as follows:

$$p_{\theta 1}(\mathbf{M}_{t-1}|\mathbf{M}_t,\mathbf{P}) = \prod_{i=1}^{N_M} \mathcal{N}(\mathbf{x}_{t-1,i}^{\mathcal{M}}; \tilde{\boldsymbol{\mu}}(\mathbf{x}_{t,i}^{\mathcal{M}},\hat{\mathbf{x}}_{0|t,i}^{\mathcal{M}}), \tilde{\beta}_t\mathbf{I}) \cdot \mathcal{C}(\mathbf{v}_{t-1,i}^{\mathcal{M}}|\tilde{\boldsymbol{c}}(\mathbf{v}_{t,i}^{\mathcal{M}},\hat{\mathbf{v}}_{0|t,i}^{\mathcal{M}})). \quad (18)$$

## B  PROOFS

### B.1  DERIVATION OF FORWARD DIFFUSION KERNELS OF OUR IPDIFF

Since our IPDIFF is designed to better generate molecular ligands that bind tightly to the given protein pockets, the prior-shifting is considered on the diffusion process of the molecular atoms' positions only. For brevity, $\mathbf{X}$ denotes the molecular atom positions, and $\mathbf{F}_0$ denotes the ground-truth interactive representations utilized in the forward process as described in Sec. 4.2.2.

Firstly, we have the marginal Gaussian for $\mathbf{X}_{t-1}$ and $\mathbf{X}_t$ as described in Equation (7):

$$q(\mathbf{X}_{t-1}|\mathbf{X}_0,\mathbf{P},\mathbf{F}_0) = \mathcal{N}(\mathbf{X}_{t-1}; \sqrt{\bar{\alpha}_{t-1}}\mathbf{X}_0 + \mathbf{S}_{t-1}, (1-\bar{\alpha}_{t-1})\boldsymbol{\Sigma}), \quad (19)$$

$$q(\mathbf{X}_t|\mathbf{X}_0,\mathbf{P},\mathbf{F}) = \mathcal{N}(\mathbf{X}_t; \sqrt{\bar{\alpha}_t}\mathbf{X}_0 + \mathbf{S}_t, (1-\bar{\alpha}_t)\boldsymbol{\Sigma}), \quad (20)$$

$$\mathbf{S}_{t-1} = \eta \cdot k_{t-1} \cdot \psi_{\theta 2}(\mathbf{F}_0, t-1), \quad \mathbf{S}_t = \eta \cdot k_t \cdot \psi_{\theta 2}(\mathbf{F}_0, t), \quad (21)$$

we can assume that:

$$q(\mathbf{X}_t|\mathbf{X}_{t-1},\mathbf{X}_0,\mathbf{P},\mathbf{F}_0) = \mathcal{N}(\mathbf{X}_t; \mathbf{A}\mathbf{X}_{t-1} + \mathbf{b}, \mathbf{L}^{-1}), \quad (22)$$

then, we can derive the marginal Gaussian for $\mathbf{X}_t$ according to Equations (19) and (22), for all $t > 1$:

$$q(\mathbf{X}_t|\mathbf{X}_0,\mathbf{P},\mathbf{F}_0) = \mathcal{N}(\mathbf{X}_t; \mathbf{A}(\sqrt{\bar{\alpha}_{t-1}}\mathbf{X}_0 + \mathbf{S}_{t-1}) + \mathbf{b}, \mathbf{L}^{-1} + (1-\bar{\alpha}_{t-1})\mathbf{A}\boldsymbol{\Sigma}\mathbf{A}^T) \quad (23)$$

$$= \mathcal{N}(\mathbf{X}_t; \sqrt{\bar{\alpha}_t}\mathbf{X}_0 + \mathbf{S}_t, (1-\bar{\alpha}_t)\boldsymbol{\Sigma}), \quad (24)$$

therefore, we can derive that:

$$\mathbf{A} = \sqrt{\alpha_t}\mathbf{I}, \quad (25)$$

$$\mathbf{b} = \mathbf{S}_t - \sqrt{\alpha_t}\mathbf{S}_{t-1}, \quad (26)$$

$$\mathbf{L}^{-1} = [1 - \bar{\alpha}_t - \alpha_t(1-\bar{\alpha}_{t-1})]\boldsymbol{\Sigma} = (1-\alpha_t)\boldsymbol{\Sigma}, \quad (27)$$

and

$$q(\mathbf{X}_t|\mathbf{X}_{t-1}, \mathbf{X}_0, \mathbf{P}, \mathbf{F}_0) = \mathcal{N}(\mathbf{X}_t; \sqrt{\alpha_t}\mathbf{X}_{t-1} + \mathbf{S}_t - \sqrt{\alpha_t}\mathbf{S}_{t-1}, \beta_t\boldsymbol{\Sigma}). \tag{28}$$

Particularly, according to Equations (19) and (28), we have:

$$q(\mathbf{X}_1|\mathbf{X}_0, \mathbf{P}, \mathbf{F}_0) = \mathcal{N}(\mathbf{X}_1; \sqrt{\alpha_1}\mathbf{X}_0 + \mathbf{S}_1 - \sqrt{\alpha_1}\mathbf{S}_0, \beta_1\boldsymbol{\Sigma}), \tag{29}$$

$$q(\mathbf{X}_2|\mathbf{X}_1, \mathbf{P}, \mathbf{F}_0) = \mathcal{N}(\mathbf{X}_2; \sqrt{\alpha_2}\mathbf{X}_1 + \mathbf{S}_2 - \sqrt{\alpha_2}\mathbf{S}_1, \beta_2\boldsymbol{\Sigma}), \tag{30}$$

$$q(\mathbf{X}_2|\mathbf{X}_0, \mathbf{P}, \mathbf{F}_0) = \mathcal{N}(\mathbf{X}_2; \sqrt{\bar{\alpha}_2}\mathbf{X}_0 + \mathbf{S}_2, (1 - \bar{\alpha}_2)\boldsymbol{\Sigma}). \tag{31}$$

from Equations (30) and (31), we can derive that:

$$q(\mathbf{X}_1|\mathbf{X}_0, \mathbf{P}, \mathbf{F}_0) = \mathcal{N}(\mathbf{X}_1; \sqrt{\alpha_1}\mathbf{X}_0 + \mathbf{S}_1, \beta_1\boldsymbol{\Sigma}), \tag{32}$$

For making Equation (29) and Equation (32) matched, we set $\mathbf{S}_0 = \mathbf{O}$.

## B.2 DERIVATION OF THE POSTERIOR DISTRIBUTIONS OF THE SHIFTED FORWARD PROCESS

Following Luo (2022), For all $t > 1$, according to the Bayes' rule:

$$q(\mathbf{X}_{t-1}|\mathbf{X}_t, \mathbf{X}_0, \mathbf{P}, \mathbf{F}_0) \tag{33}$$

$$= \frac{q(\mathbf{X}_t|\mathbf{X}_{t-1}, \mathbf{X}_0, \mathbf{P}, \mathbf{F}_0)q(\mathbf{X}_{t-1}|\mathbf{X}_0, \mathbf{P}, \mathbf{F}_0)}{q(\mathbf{X}_t|\mathbf{X}_0, \mathbf{P}, \mathbf{F}_0)} \tag{34}$$

$$= \frac{\mathcal{N}(\mathbf{X}_t; \sqrt{\alpha_t}\mathbf{X}_{t-1} + \mathbf{S}_t - \sqrt{\alpha_t}\mathbf{S}_{t-1}, \beta_t\boldsymbol{\Sigma}) \cdot \mathcal{N}(\mathbf{X}_{t-1}; \sqrt{\bar{\alpha}_{t-1}}\mathbf{X}_0 + \mathbf{S}_{t-1}, (1 - \bar{\alpha}_{t-1})\boldsymbol{\Sigma})}{\mathcal{N}(\mathbf{X}_t; \sqrt{\bar{\alpha}_t}\mathbf{X}_0 + \mathbf{S}_t, (1 - \bar{\alpha}_t)\boldsymbol{\Sigma})} \tag{35}$$

$$\propto \exp\{-\frac{1}{2}[\frac{(\mathbf{X}_t - \sqrt{\alpha_t}\mathbf{X}_{t-1} - \mathbf{S}_t + \sqrt{\alpha_t}\mathbf{S}_{t-1})^2}{(1 - \alpha_t)} + \frac{(\mathbf{X}_{t-1} - \sqrt{\bar{\alpha}_{t-1}}\mathbf{X}_0 - \mathbf{S}_{t-1})^2}{(1 - \bar{\alpha}_{t-1})}$$

$$- \frac{(\mathbf{X}_t - \sqrt{\bar{\alpha}_t}\mathbf{X}_0 - \mathbf{S}_t)^2}{(1 - \bar{\alpha}_t)}]\} \tag{36}$$

$$= \exp\{-\frac{1}{2}[\frac{\alpha_t\mathbf{X}_{t-1}^2 - 2\sqrt{\alpha_t}\mathbf{X}_t\mathbf{X}_{t-1} + 2\sqrt{\alpha_t}\mathbf{X}_{t-1}\mathbf{S}_t - \alpha_t\mathbf{X}_{t-1}\mathbf{S}_{t-1}}{1 - \alpha_t}$$

$$+ \frac{\mathbf{X}_{t-1}^2 - 2\sqrt{\bar{\alpha}_{t-1}}\mathbf{X}_{t-1}\mathbf{X}_0 - 2\mathbf{X}_{t-1}\mathbf{S}_{t-1}}{1 - \bar{\alpha}_{t-1}}]\} + C(\mathbf{X}_0, \mathbf{X}_t) \tag{37}$$

$$\propto \exp\{-\frac{1}{2}[(\frac{\alpha_t}{1 - \alpha_t} + \frac{1}{1 - \bar{\alpha}_{t-1}})\mathbf{X}_{t-1}^2 + 2(\frac{-\sqrt{\alpha_t}\mathbf{X}_t + \sqrt{\alpha_t}\mathbf{S}_t - \alpha_t\mathbf{S}_{t-1}}{1 - \alpha_t}$$

$$- \frac{\sqrt{\bar{\alpha}_{t-1}}\mathbf{X}_0 + \mathbf{S}_{t-1}}{1 - \bar{\alpha}_{t-1}})\mathbf{X}_{t-1}]\} \tag{38}$$

$$= \exp\{-\frac{1}{2}(\frac{1 - \bar{\alpha}_t}{(1 - \alpha_t)(1 - \bar{\alpha}_{t-1})})[\mathbf{X}_{t-1}^2$$

$$+ 2(\frac{\frac{-\sqrt{\alpha_t}\mathbf{X}_t + \sqrt{\alpha_t}\mathbf{S}_t - \alpha_t\mathbf{S}_{t-1}}{1 - \alpha_t} + \frac{\sqrt{\bar{\alpha}_{t-1}}\mathbf{X}_0 + \mathbf{S}_{t-1}}{1 - \bar{\alpha}_{t-1}}}{\frac{1 - \bar{\alpha}_t}{(1 - \alpha_t)(1 - \bar{\alpha}_{t-1})}})\mathbf{X}_{t-1}]\} \tag{39}$$

$$= \exp\{-\frac{1}{2}(\frac{1 - \bar{\alpha}_t}{(1 - \alpha_t)(1 - \bar{\alpha}_{t-1})})[\mathbf{X}_{t-1}^2$$

$$+ 2\frac{(-\sqrt{\alpha_t}\mathbf{X}_t + \sqrt{\alpha_t}\mathbf{S}_t - \alpha_t\mathbf{S}_{t-1})(1 - \bar{\alpha}_{t-1}) - (\sqrt{\bar{\alpha}_{t-1}}\mathbf{X}_0 + \mathbf{S}_{t-1})(1 - \alpha_t)}{1 - \bar{\alpha}_t}\mathbf{X}_{t-1}]\} \tag{40}$$

$$= \exp\{-\frac{1}{2}(\frac{1}{\frac{1 - \bar{\alpha}_t}{(1 - \alpha_t)(1 - \bar{\alpha}_{t-1})}})[\mathbf{X}_{t-1}^2$$

$$- 2(\frac{\sqrt{\bar{\alpha}_{t-1}}\beta_t}{1 - \bar{\alpha}_t}\mathbf{X}_0 + \frac{\sqrt{\alpha_t}(1 - \bar{\alpha}_{t-1})}{1 - \bar{\alpha}_t}\mathbf{X}_t - \sqrt{\alpha_t}\frac{1 - \bar{\alpha}_{t-1}}{1 - \bar{\alpha}_t}\mathbf{S}_t + \mathbf{S}_{t-1})\mathbf{X}_{t-1}]\} \tag{41}$$

$$\propto \mathcal{N}(\mathbf{X}_{t-1}; \frac{\sqrt{\bar{\alpha}_{t-1}}\beta_t}{1 - \bar{\alpha}_t}\mathbf{X}_0 + \frac{\sqrt{\alpha_t}(1 - \bar{\alpha}_{t-1})}{1 - \bar{\alpha}_t}\mathbf{X}_t - \sqrt{\alpha_t}\frac{1 - \bar{\alpha}_{t-1}}{1 - \bar{\alpha}_t}\mathbf{S}_t + \mathbf{S}_{t-1}, \frac{(1 - \bar{\alpha}_{t-1})}{1 - \bar{\alpha}_t}\beta_t\boldsymbol{\Sigma})$$

$$\tag{42}$$

As mentioned in Sec. 4.2.1, since the ground-truth molecule $\mathbf{X}_0^{\mathcal{M}}$ and $\mathbf{V}_0^{\mathcal{M}}$ are inaccessible at the time step $t$ of the reverse process, we utilize the molecule $\hat{\mathbf{M}}_{0|t+1} = [\hat{\mathbf{X}}_{0|t+1}^{\mathcal{M}}, \hat{\mathbf{V}}_{0|t+1}^{\mathcal{M}}]$ estimated in the previous time step $t+1$ to substitute the $\mathbf{M}_0$ and feed it with $\mathbf{P}$ into the pretrained IPNET for obtaining the interactive representation $\mathbf{F}_{0|t+1}^{\mathcal{M}}$. And then we calculate the $\hat{\mathbf{S}}_t$ from $\mathbf{F}_{0|t+1}^{\mathcal{M}}$ according to the Equation (6). Similarly, we calculate the $\hat{\mathbf{S}}_{t-1}$ from $\mathbf{F}_{0|t}^{\mathcal{M}}$. In IPDIFF, we employ a model $\phi_{\theta 1}(\mathbf{X}_t, t)$ for predicting $\mathbf{X}_0$ directly. Then we can get the predicted posterior distributions parameterized by $\theta 1$:

$$
\begin{aligned}
p_{\theta 1}(\mathbf{X}_{t-1}|\mathbf{X}_t, \mathbf{F}_{0|t+1}, \mathbf{F}_{0|t}) =& \mathcal{N}(\frac{\sqrt{\bar{\alpha}_{t-1}}\beta_t}{1-\bar{\alpha}_t}\phi_{\theta 1}(\mathbf{X}_t, t) + \frac{\sqrt{\bar{\alpha}_t}(1-\bar{\alpha}_{t-1})}{1-\bar{\alpha}_t}\mathbf{X}_t \\
& - \sqrt{\bar{\alpha}_t}\frac{1-\bar{\alpha}_{t-1}}{1-\bar{\alpha}_t}\hat{\mathbf{S}}_t + \hat{\mathbf{S}}_{t-1}, \frac{(1-\bar{\alpha}_{t-1})}{1-\bar{\alpha}_t}\beta_t\mathbf{\Sigma}).
\end{aligned}
\tag{43}
$$

### B.3 DERIVATION OF THE TRAINING OBJECTIVES

According to Equation (42), the training objective can be represented as:

$$
\begin{aligned}
L =& \mathbb{E}_q\{-log p_{\theta 1}(\mathbf{X}_0|\mathbf{X}_1, \mathbf{F}_{0|t+1}, \mathbf{F}_{0|t}) + \mathcal{D}_{KL}[q(\mathbf{X}_T|\mathbf{X}_0, \mathbf{F}_0)\|p_{\theta 1}(\mathbf{X}_T)] \\
& + \sum_{t=2}^T \mathcal{D}_{KL}[q(\mathbf{X}_{t-1}|\mathbf{X}_t, \mathbf{X}_0, \mathbf{F}_0)\|p_{\theta 1}(\mathbf{X}_{t-1}|\mathbf{X}_t, \mathbf{F}_{0|t+1}, \mathbf{F}_{0|t})]\}
\end{aligned}
\tag{44}
$$

For the first and the second terms, we can derive them as constants $c$ and discard them in the objective function. For the third term, we can derive it by Gaussian Keullback-Leibler divergence:

$$
\begin{aligned}
& D_{KL}[q(\mathbf{X}_{t-1}|\mathbf{X}_t, \mathbf{X}_0, \mathbf{F}_0)\|p_{\theta 1}(\mathbf{X}_{t-1}|\mathbf{X}_t, \mathbf{F}_{0|t+1}, \mathbf{F}_{0|t})] \\
=& \frac{1}{2}\|\frac{\sqrt{\bar{\alpha}_{t-1}}\beta_t}{1-\bar{\alpha}_t}[\phi_{\theta 1}(\mathbf{X}_t, t) - \mathbf{X}_0] + \sqrt{\bar{\alpha}_t}\frac{1-\bar{\alpha}_{t-1}}{1-\bar{\alpha}_t}(\hat{\mathbf{S}}_t - \mathbf{S}_t) + (\hat{\mathbf{S}}_{t-1} - \mathbf{S}_{t-1})\|^2_{(\frac{1-\bar{\alpha}_{t-1}}{1-\bar{\alpha}_t}\beta_t\mathbf{\Sigma})^{-1}}.
\end{aligned}
\tag{45}
$$

Assuming that $\mathbf{S}_t$ is Lipschitz continuous w.r.t $\mathbf{X}_0$, then we can simplify the Gaussian Keullback-Leibler divergence:

$$
\begin{aligned}
& \frac{1}{2}\|\frac{\sqrt{\bar{\alpha}_{t-1}}\beta_t}{1-\bar{\alpha}_t}[\phi_{\theta 1}(\mathbf{X}_t, t) - \mathbf{X}_0] + \sqrt{\bar{\alpha}_t}\frac{1-\bar{\alpha}_{t-1}}{1-\bar{\alpha}_t}(\hat{\mathbf{S}}_t - \mathbf{S}_t) + (\hat{\mathbf{S}}_{t-1} - \mathbf{S}_{t-1})\|_{(\frac{1-\bar{\alpha}_{t-1}}{1-\bar{\alpha}_t}\beta_t\mathbf{\Sigma})^{-1}} \\
\leq& c_t\|[\phi_{\theta 1}(\mathbf{X}_t, t) - \mathbf{X}_0] + \sqrt{\bar{\alpha}_t}\frac{1-\bar{\alpha}_{t-1}}{1-\bar{\alpha}_t}(\hat{\mathbf{S}}_t - \mathbf{S}_t) + (\hat{\mathbf{S}}_{t-1} - \mathbf{S}_{t-1})\| \\
\leq& c_t(\|[\phi_{\theta 1}(\mathbf{X}_t, t) - \mathbf{X}_0]\| + \|\sqrt{\bar{\alpha}_t}\frac{1-\bar{\alpha}_{t-1}}{1-\bar{\alpha}_t}(\hat{\mathbf{S}}_t - \mathbf{S}_t)\| + \|(\hat{\mathbf{S}}_{t-1} - \mathbf{S}_{t-1})\|) \\
\leq& \gamma_t\|[\phi_{\theta 1}(\mathbf{X}_t, t) - \mathbf{X}_0]\|,
\end{aligned}
\tag{46}
$$

by the lipschitz continuity of Mahalanobis Distances and $\mathbf{S}_t$. Here $c_t, \gamma_t$ are scaling factors. Finally, the training objective of atom position at time step $t-1$ are defined as follows:

$$
L_{t-1}^{(x)} = \frac{1}{2\tilde{\beta}_t^2}\sum_{i=1}^{N_M}\|\tilde{\boldsymbol{\mu}}(\mathbf{x}_{t,i}, \mathbf{x}_{0,i}, \mathbf{f}_{0,i}) - \tilde{\boldsymbol{\mu}}(\mathbf{x}_{t,i}, \hat{\mathbf{x}}_{0,i}, \mathbf{f}_{0|t+1,i}, \mathbf{f}_{0|t,i})\|^2 = \gamma_t\sum_{i=1}^{N_M}\|\mathbf{x}_{0,i} - \hat{\mathbf{x}}_{0,i}\|; \tag{47}
$$

where $\hat{\mathbf{X}}_0$ and $\hat{\mathbf{V}}_0$ are predicted from $\mathbf{X}_t$ and $\mathbf{V}_t$, where $\gamma_t$ is a scaling factor. And we use the same objective function of atom type at time step $t-1$ as Guan et al. (2023a):

$$
L_{t-1}^{(v)} = \sum_{i=1}^{N_M}\sum_{k=1}^K \tilde{\boldsymbol{c}}(\mathbf{v}_{t,i}, \mathbf{v}_{0,i})_k \log \frac{\tilde{\boldsymbol{c}}(\mathbf{v}_{t,i}, \mathbf{v}_{0,i})_k}{\tilde{\boldsymbol{c}}(\mathbf{v}_{t,i}, \hat{\mathbf{v}}_{0,i})_k}; . \tag{48}
$$

Kindly recall that $\mathbf{x}_{t,i}, \mathbf{v}_{t,i}, \hat{\mathbf{x}}_{0,i}, \hat{\mathbf{v}}_{0,i}, \mathbf{f}_{0,i}, \mathbf{f}_{0|t+1,i}$ and $\mathbf{f}_{0|t,i}$ correspond to the $i$-th row of $\mathbf{X}_t, \mathbf{V}_t, \hat{\mathbf{X}}_0, \hat{\mathbf{V}}_0, \mathbf{F}_{0,i}, \mathbf{F}_{0|t+1,i}$ and $\mathbf{F}_{0|t,i}$, respectively. The final loss combines the above two losses with a hyperparameter $\lambda$ as: $L = L_{t-1}^{(x)} + \lambda L_{t-1}^{(v)}$. We summarize the training procedure of IPDIFF in Algorithm 1 and highlight the differences from its counterpart, TargetDiff (Guan et al., 2023a), in blue.

### B.4 THE BETTER LIKELIHOOD

We shall show that IPDIFF is theoretically able to achieve better likelihood compared to previous diffusion models. As the exact likelihood is intractable, we aim to compare the optimal variational bounds for negative log likelihoods (NLL). The objective function of IPDIFF at time step t is

$$E_{q_{\theta_2}} D_{KL}(q_{\theta_2}(x_{t-1}|x_t, x_0, F_0, P)||p_{\theta_1}(x_{t-1}|x_t, F_{0|t+1}, F_{0|t}, P)) \tag{49}$$

and its optimal solution is

$$\min_{\theta_1, \theta_2} E_{q_{\theta_2}} D_{KL}(q_{\theta_2}(x_{t-1}|x_t, x_0, F_0, P)||p_{\theta_1}(x_{t-1}|x_t, F_{0|t+1}, F_{0|t}, P)) \tag{50}$$

$$= min_{\theta_2}[min_{\theta_1} E_{q_{\theta_2}} D_{KL}(q_{\theta_2}(x_{t-1}|x_t, x_0, F_0, P)||p_{\theta_1}(x_{t-1}|x_t, F_{0|t+1}, F_{0|t}, P)) \tag{51}$$

$$\leq min_{\theta_1} E_q D_{KL}(q(x_{t-1}|x_t, x_0, P)||p_{\theta_1}(x_{t-1}|x_t, P)), \tag{52}$$

where $min_{\theta_1} E_q D_{KL}(q(x_{t-1}|x_t, x_0, P)||p_{\theta_1}(x_{t-1}|x_t, P))$ is the optimal loss of previous diffusion models that do not use interactive representations $F$ in the forward process. Similar inequality can be obtained for t=1:

$$\min_{\theta_1, \theta_2} E_{q_{\theta_2}} - \log p_{\theta_1}(x_0|x_1, F_{0|1}, P) \tag{53}$$

$$\leq \min_{\theta_1} E_q - \log p_{\theta_1}(x_0|x_1, P) \tag{54}$$

As a result, we have the following inequality by summing up the objectives at all time steps:

$$- E_{q(x_0|P)} \log p_{\theta_1}(x_0|P) \tag{55}$$

$$\leq min_{\theta_1, \theta_2} \sum_{t>1} E_{q_{\theta_2}} D_{KL}(q_{\theta_2}(x_{t-1}|x_t, x_0, F_0, P)||p_{\theta_1}(x_{t-1}|x_t, F_{0|t+1}, F_{0|t}, P)) \tag{56}$$

$$+ E_{q_{\theta_2}} - \log p_{\theta_1}(x_0|x_1, F_{0|1}, P) + C$$

$$\leq min_{\theta_1} \sum_{t>1} E_q D_{KL}(q(x_{t-1}|x_t, x_0, P)||p_{\theta_1}(x_{t-1}|x_t, P)) + E_q - \log p_{\theta_1}(x_0|x_1, P) + C \tag{57}$$

where C is a constant defined by $\sqrt{\bar{\alpha}_T}$. Hence, IPDIFF has a tighter bound for the NLL, and thus theoretically capable of achieving better likelihood, compared with the previous diffusion models.

## C TRAINING AND SAMPLING PROCEDURE

We summarize the training and sampling procedure as Algorithms 1 and 2.

## D IMPLEMENTATION DETAILS

### D.1 DETAILS OF IPNET

**Initialization of Inputs** Following (Guan et al., 2023a), we use a one-hot element indicator {H, C, N, O, S, Se} and one-hot amino acid type indicator (20 types) to represent each protein atom. Similarly, each ligand atom are repsented with a one-hot element indicator {C, N, O, F, P, S, Cl}. And an additional one-dimensional flag indicating whether the atoms belong to the protein or ligand are introduced. Two 1-layer MLPs are used to map the input protein and ligand into 128-dim latent spaces respectively.

**Architectures** The IPNET is designed to model the complex intramolecular and intermolecular 3D interactions between the atoms of proteins-ligand pairs. To achieve this, we use three shallow SE(3)-equivariant neural networks for geometric message passing on the fully-connected graphs of the protein, ligand and complexes (consists of protein and ligand), respectively. We then apply a cross attention layer to the paired protein-ligand graph for learning the inter-molecule interactions. Finally, we use a sum-pooling layer to extract a global representation of the protein-ligand pair by pooling all atom nodes. And a two-layer MLP is introduced to predict the binding affinity $S_{\text{Aff}}$. More details about the model architecture are provided in Tab. 6.

---

**Algorithm 1** Training Procedure of IPDIFF

---

**Input:** Protein-ligand binding dataset $\{\mathcal{P}, \mathcal{M}\}_{i=1}^N$, learnable diffusion denoising model $\phi_{\theta 1}$, learnable neural network $\psi_{\theta 2}$ and pretrained interaction prior network IPNET

1: **while** $\phi_{\theta 1}$ and $\psi_{\theta 2}$ not converge **do**
2: $\quad \llbracket \mathbf{X}_0^{\mathcal{M}}, \mathbf{X}_0^{\mathcal{P}} \rrbracket, \llbracket \mathbf{V}_0^{\mathcal{M}}, \mathbf{V}_0^{\mathcal{P}} \rrbracket \sim \{\mathcal{P}, \mathcal{M}\}_{i=1}^N$
3: $\quad t \sim \mathcal{U}(0, \dots, T)$
4: $\quad$ Move the complex to make CoM of protein atoms zero
5: $\quad$ Obtain interactive features $\llbracket \mathbf{F}_0^{\mathcal{M}}, \mathbf{F}_0^{\mathcal{P}} \rrbracket$ from IPNET:
$\qquad \llbracket \mathbf{F}_0^{\mathcal{M}}, \mathbf{F}_0^{\mathcal{P}} \rrbracket = \text{IPNET}(\llbracket \mathbf{X}_0^{\mathcal{M}}, \mathbf{X}_0^{\mathcal{P}} \rrbracket, \llbracket \mathbf{V}_0^{\mathcal{M}}, \mathbf{V}_0^{\mathcal{P}} \rrbracket)$
6: $\quad$ Perturb $\mathbf{X}_0^{\mathcal{M}}$ to obtain $\mathbf{X}_t^{\mathcal{M}}$ with shifts $\mathbf{S}_t^{\mathcal{M}}$:
$\qquad \epsilon \sim \mathcal{N}(0, \boldsymbol{I})$
$\qquad \mathbf{S}_t^{\mathcal{M}} = \eta \cdot k_t \cdot \psi_{\theta 2}(\mathbf{F}_0^{\mathcal{M}}, t)$ $\hfill$ (Equation (6))
$\qquad \mathbf{X}_t^{\mathcal{M}} = \sqrt{\bar{\alpha}_t} \mathbf{X}_0^{\mathcal{M}} + \mathbf{S}_t^{\mathcal{M}} + \sqrt{1 - \bar{\alpha}_t} \epsilon$
7: $\quad$ Perturb $\mathbf{V}_0^{\mathcal{M}}$ to obtain $\mathbf{V}_t^{\mathcal{M}}$:
$\qquad g \sim Gumbel(0, 1)$
$\qquad \log \mathbf{c}^{\mathcal{M}} = \log(\bar{\alpha}_t \mathbf{V}_0^{\mathcal{M}} + (1 - \bar{\alpha}_t / K))$
$\qquad \mathbf{V}_t^{\mathcal{M}} = onehot(\arg\max_i (g_i + \log c_i^{\mathcal{M}}))$
8: $\quad$ Embed $\mathbf{V}_t^{\mathcal{M}}$ into $\tilde{\mathbf{H}}_t^{\mathcal{M},0}$, and embed $\mathbf{V}_0^{\mathcal{P}}$ into $\tilde{\mathbf{H}}_t^{\mathcal{P},0}$ ($\tilde{\mathbf{H}}_0^{\mathcal{P},0} = \cdots = \tilde{\mathbf{H}}_T^{\mathcal{P},0}$)
9: $\quad$ Obtain features $\llbracket \mathbf{H}_t^{\mathcal{M},0}, \mathbf{H}_t^{\mathcal{P},0} \rrbracket$ through prior-conditioning:
$\qquad \llbracket \mathbf{H}_t^{\mathcal{M},0}, \mathbf{H}_t^{\mathcal{P},0} \rrbracket = concat(\llbracket \tilde{\mathbf{H}}_t^{\mathcal{M}}, \tilde{\mathbf{H}}_t^{\mathcal{P}} \rrbracket, \llbracket \mathbf{F}_0^{\mathcal{M}}, \mathbf{F}_0^{\mathcal{P}} \rrbracket)$ $\hfill$ (Equation (12))
10: $\quad$ Predict $(\hat{\mathbf{X}}_{0|t}^{\mathcal{M}}, \hat{\mathbf{V}}_{0|t}^{\mathcal{M}})$ from $\phi_{\theta 1}$:
$\qquad \hat{\mathbf{X}}_{0|t}^{\mathcal{M}}, \hat{\mathbf{V}}_{0|t}^{\mathcal{M}} = \phi_{\theta 1}(\llbracket \mathbf{X}_t^{\mathcal{M}}, \mathbf{X}_0^{\mathcal{P}} \rrbracket, \llbracket \mathbf{H}_t^{\mathcal{M},0}, \mathbf{H}_t^{\mathcal{P},0} \rrbracket)$ $\hfill$ (Equations (13) and (14))
11: $\quad$ Compute loss $L$ with $(\hat{\mathbf{X}}_{0|t}^{\mathcal{M}}, \hat{\mathbf{V}}_{0|t}^{\mathcal{M}})$ and $(\mathbf{X}_0^{\mathcal{M}}, \mathbf{V}_0^{\mathcal{M}})$ $\hfill$ (Equations (47) and (48))
12: $\quad$ Jointly update $\theta 1$ and $\theta 2$ by minimizing $L$
13: **end while**

---

**Training Details** During the training, we use the Mean Squared Error (MSE) loss with respect to the difference between the predicted and ground truth binding affinity scores as the optimization objective. The binding affinity values of protein-ligand pairs range from 2.0 to 11.92. For avoiding information leakage, we filter the training set by calculating the Tanimoto similarity with the molecules in the testing set of CrossDocked2020, and the similarity threshold was set to 0.1. As a result, there are 23 complexes filtered out from the training set. We train IPNET on a single NVIDIA V100 GPU, and we use the Adam as our optimizer with learning rate 0.001, $betas = (0.95, 0.999)$, batch size 16. The experiments are conducted on PDBBind v2016 dataset as mentioned in Sec. 5.1.

**Evaluation of IPNET** Following Li et al. (2021), we select Root Mean Square Error (RMSE), Mean Absolute Error (MAE), Pearson's correlation coefficient (R) and the standard deviation (SD) in regression to measure the prediction error. Meanwhile, we use these metrics to select the pretrained IPNET utilized in IPDIFF because we believe that the ability to predict the binding affinity is highly related to the interaction modeling. The testing results of IPNET (in Tabs. 3 and 8) in binding affinity prediction are present in Tab. 5. And we introduce two GNN-based binding affinity prediction methods: GraphDTA Nguyen et al. (2021) and GNN-DTI (Lim et al., 2019) to make comparisons, indicating the rationality of our model design.

Table 5: Performance of IPNET in binding affinity prediction. ($\uparrow$) / ($\downarrow$) denotes a larger / smaller number is better. Top 2 results are highlighted with **bold text** and underlined text, respectively.

| Methods | RMSE ($\downarrow$) | MAE ($\downarrow$) | SD ($\downarrow$) | R ($\uparrow$) |
|---|---|---|---|---|
| GraphDTA | 1.562 | 1.191 | 1.558 | 0.697 |
| GNN-DTI | 1.492 | 1.192 | 1.471 | 0.736 |
| IPNET-Seq. | 1.566 | 1.263 | 1.547 | 0.704 |
| IPNET ($\sigma = 0$) | 1.612 | 1.317 | 1.589 | 0.683 |
| IPNET ($\sigma = 0.1$) | 1.544 | 1.239 | 1.516 | 0.718 |
| IPNET ($\sigma = 0.5$) | **1.439** | **1.140** | **1.386** | **0.771** |
| IPNET ($\sigma = 1$) | 1.641 | 1.332 | 1.590 | 0.683 |

---

**Algorithm 2** Sampling Procedure of IPDIFF

---

**Input:** The protein binding site $\mathcal{P}$, the learned diffusion denoising model $\phi_{\theta 1}$, the learned neural network $\psi_{\theta 2}$ and pretrained interaction prior network IPNET

**Output:** Generated ligand molecule $\mathcal{M}$ that binds to the protein pocket $\mathcal{P}$

1: Sample the number of atoms $N_M$ of the ligand molecule $\mathcal{M}$ as described in Sec. 3
2: Move CoM of protein atoms to zero
3: Sample initial ligand atom coordinates $\mathbf{X}_T^{\mathcal{M}}$ and atom types $\mathbf{V}_T^{\mathcal{M}}$
4: Let $[\![\mathbf{F}_{0|T+1}^{\mathcal{M}}, \mathbf{F}_{0|T+1}^{\mathcal{P}}]\!] = \mathbf{O}, \mathbf{S}_T^{\mathcal{M}} = \mathbf{O}$
5: **for** $t$ in $T, \ldots, 1$ **do**
6:     Embed $\mathbf{V}_t^{\mathcal{M}}$ into $\tilde{\mathbf{H}}_t^{\mathcal{M},0}$, and embed $\mathbf{V}_0^{\mathcal{P}}$ into $\tilde{\mathbf{H}}_t^{\mathcal{P},0}$ $(\tilde{\mathbf{H}}_0^{\mathcal{P},0} = \cdots = \tilde{\mathbf{H}}_T^{\mathcal{P},0})$
7:     Obtain features $[\![\mathbf{H}_t^{\mathcal{M},0}, \mathbf{H}_t^{\mathcal{P},0}]\!]$ through prior-conditioning:
        $[\![\mathbf{H}_t^{\mathcal{M},0}, \mathbf{H}_t^{\mathcal{P},0}]\!] = \text{concat}([\![\tilde{\mathbf{H}}_t^{\mathcal{M}}, \tilde{\mathbf{H}}_t^{\mathcal{P}}]\!], [\![\mathbf{F}_{0|t+1}^{\mathcal{M}}, \mathbf{F}_{0|t+1}^{\mathcal{P}}]\!])$     (Equation (12))
8:     Predict $(\hat{\mathbf{X}}_{0|t}^{\mathcal{M}}, \hat{\mathbf{V}}_{0|t}^{\mathcal{M}})$ from $\phi_{\theta 1}$:
        $\hat{\mathbf{X}}_{0|t}^{\mathcal{M}}, \hat{\mathbf{V}}_{0|t}^{\mathcal{M}} = \phi_{\theta 1}([\![\mathbf{X}_t^{\mathcal{M}}, \mathbf{X}_0^{\mathcal{P}}]\!], [\![\mathbf{H}_t^{\mathcal{M},0}, \mathbf{H}_t^{\mathcal{P},0}]\!])$     (Equations (13) and (14))
9:     Sample $\mathbf{X}_{t-1}^{\mathcal{M}}$ from the shifted posterior $p_{\theta 1}(\mathbf{X}_{t-1}^{\mathcal{M}}|\mathbf{X}_t^{\mathcal{M}}, \mathbf{X}_0^{\mathcal{P}}, \mathbf{F}_{0|t+1}^{\mathcal{M}})$:     (Equation (11))
        where $z \sim \mathcal{N}(0, \boldsymbol{I})$
        $\mathbf{S}_{t-1}^{\mathcal{M}} = \eta \cdot k_{t-1} \cdot \psi_{\theta 2}(\mathbf{F}_{0|t+1}^{\mathcal{M}}, t-1)$     (Equation (6))
        $\mathbf{X}_{t-1}^{\mathcal{M}} = \frac{\sqrt{\bar{\alpha}_{t-1}}\beta_t}{1-\bar{\alpha}_t}\hat{\mathbf{X}}_{0|t}^{\mathcal{M}} + \frac{\sqrt{\alpha_t}(1-\bar{\alpha}_{t-1})}{1-\bar{\alpha}_t}(\mathbf{X}_t^{\mathcal{M}} - \mathbf{S}_t^{\mathcal{M}}) + \mathbf{S}_{t-1}^{\mathcal{M}} + \sqrt{\frac{1-\bar{\alpha}_{t-1}}{1-\bar{\alpha}_t}\beta_t}z$
10:    Sample $\mathbf{V}_{t-1}^{\mathcal{M}}$ from the posterior $q(\mathbf{V}_{t-1}^{\mathcal{M}}|\hat{\mathbf{V}}_{0|t}^{\mathcal{M}}, \mathbf{V}_t^{\mathcal{M}}, \mathbf{V}_0^{\mathcal{P}})$     (Equation (18))
11:    Obtain features $[\![\mathbf{F}_{0|t}^{\mathcal{M}}, \mathbf{F}_{0|t}^{\mathcal{P}}]\!]$ through IPNET:
        $[\![\mathbf{F}_{0|t}^{\mathcal{M}}, \mathbf{F}_{0|t}^{\mathcal{P}}]\!] = \text{IPNET}([\![\hat{\mathbf{X}}_{0|t}^{\mathcal{M}}, \mathbf{X}_0^{\mathcal{P}}]\!], [\![\hat{\mathbf{V}}_{0|t}^{\mathcal{M}}, \mathbf{V}_0^{\mathcal{P}}]\!])$
12: **end for**

---

## D.2 DETAILS OF OUR PRIOR-GUIDED DIFFUSION MODEL

**Initialization of Inputs**   The representation of the atoms in the proteins and molecules is the same as the representation used in IPNET Appendix D.1. A 4-dim one-hot vector indicating four bond types: bond between protein atoms, ligand atoms, protein-ligand atoms or ligand-protein atoms is introduced to representing the connection between atoms. And we introduce distance embeddings by using the distance with radial basis functions located at 20 centers between 0 Å and 10 Å. Finally we calculate the outer products of distance embedding and bond types to obtain the edge features.

**Architectures**   At the $l$-th layer, we dynamically construct the protein-ligand complex with a $k$-nearest neighbors (knn) graph based on coordinates of the given protein and the ligand from previous layer. In practice, we set the number of neighbors $k_n = 32$. As mentioned in Sec. 4.1, we apply an SE(3)-equivariant neural network for message passing. The 9-layer equivariant neural network consists of Transformer layers with 128-dim hidden layer and 16 attention heads. Following Guan et al. (2023a), in the diffusion process, we select the fixed sigmoid $\beta$ schedule with $\beta_1 = 1\mathrm{e}{-7}$ and $\beta_T = 2\mathrm{e}{-3}$ as variance schedule for atom coordinates, and the cosine $\beta$ schedule with $s = 0.01$ for atom types. The number of diffusion steps are set to 1000. We denote PCM and PSM as prior-conditioning and prior-shifting mechanisms, repecitively. More details about the model architecture are provided in Tab. 6.

**Training Details**   Following Guan et al. (2023a), we use the Adam as our optimizer with learning rate 0.001, $betas = (0.95, 0.999)$, batch size 4 and clipped gradient norm 8. We balance the atom type loss and atom position loss by multiplying a scaling factor $\lambda = 100$ on the atom type loss. During the training phase, we add a small Gaussian noise with a standard deviation of 0.1 to protein atom coordinates as data augmentation. We train the parameterized diffusion denoising model of our IPDIFF on a single NVIDIA V100 GPU, and it could converge within 200k steps.

Table 6: Details of both IPNET and Prior-Guided Diffusion Model in our IPDIFF

| Network | Module | Backbone | Input Dimensions | Output Dimensions | Blocks |
|---|---|---|---|---|---|
| IPNET | Protein Encoder | EGNN | $N_P \times 128$ | $N_P \times 128$ | 2 |
| | Ligand Encoder | EGNN | $N_M \times 128$ | $N_M \times 128$ | 2 |
| | Complex Encoder | EGNN | $(N_P + N_M) \times 128$ | $(N_P + N_M) \times 128$ | 2 |
| | Interaction Layer | Graph Attention Layer | $(N_P + N_M) \times 128$ | $(N_P + N_M) \times 128$ | 1 |
| | Pooling | Sum-pooling | $(N_P + N_M) \times 128$ | $1 \times 128$ | 1 |
| IPNET-Seq. | Protein Encoder | Graph Attention Layer | $N_P \times 128$ | $N_P \times 128$ | 2 |
| | Ligand Encoder | Graph Attention Layer | $N_M \times 128$ | $N_M \times 128$ | 2 |
| | Complex Encoder | Graph Attention Layer | $(N_P + N_M) \times 128$ | $(N_P + N_M) \times 128$ | 2 |
| | Interaction Layer | Graph Attention Layer | $(N_P + N_M) \times 128$ | $(N_P + N_M) \times 128$ | 1 |
| | Pooling | Sum-pooling | $(N_P + N_M) \times 128$ | $1 \times 128$ | 1 |
| Diffusion Denoising Network | Position Dynamics | Transformer | $(N_P + N_M) \times 3$ | $(N_P + N_M) \times 3$ | 9 |
| | Atom Type Dynamics | Transformer | $(N_P + N_M) \times 128$ | $(N_P + N_M) \times 128$ | 9 |
| | PCM Fusion Layer | MLP | $(N_P + N_M) \times (128 + 128)$ | $(N_P + N_M) \times 128$ | 1 |
| | PSM Adapter Layer | MLP | $N_M \times (128 + 1)$ | $N_M \times 3$ | 1 |

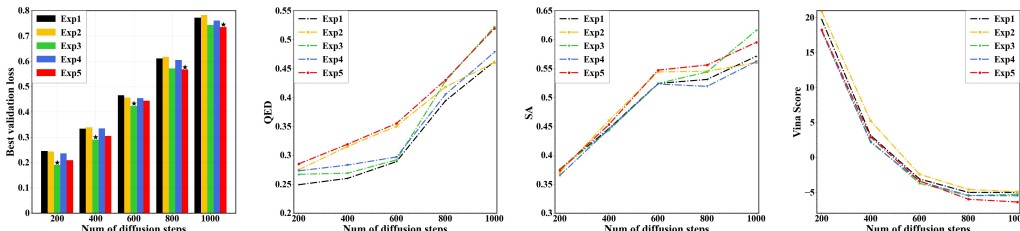

Figure 5: Ablation study on diffusion step number. We compare 5 experiments described in Sec. 5.3 in terms of best validation loss, QED, SA and Vina Score under different diffusion steps settings.

# E    MORE ABLATION STUDIES

**Effect of Diffusion Steps**    We conduct a set of experiments: (1) **Exp1**: the baseline model which is obtained by removing prior-conditioning and prior-shifting mechanisms from IPDIFF; (2) **Exp2**: the baseline model equipped with the self-conditioning mechanism which can be regarded as our prior-conditioning mechanism without IPNET, (3) **Exp3**: the baseline model equipped with the prior-conditioning mechanism as mentioned in Sec. 4.2.2, (4) **Exp4**: the baseline model equipped with the prior-shifting mechanism as mentioned in Sec. 4.2.1, (5) **Exp5**: our IPDIFF, for comparing the performance of **Exp1-5** under different number of diffusion steps. We firstly train the models with 200, 400, 600, 800, 1000 diffusion steps, and then evaluate the generated molecules by sampling the same number of steps as training. As shown in Figure 5, the baseline model equipped with prior-conditioning and prior-shifting mechanism can achieve better validation loss under each setting of diffusion steps. Under the same setting, the validation loss can be viewed as a surrogate of negative Evidence Lower Bound (ELBO) and lower validation loss means the model can better approximate the data distribution (Guan et al., 2023b). The fact that the model trained with fewer diffusion steps achieves lower validation loss is because it fits noises better at fewer time steps with limited model capacity. Moreover, the comparison results demonstrate that our model equipped with prior-conditioning and prior-shifting mechanism is able to generate high-quality ligand molecules (high QED and SA, low Vina) even with fewer sampling steps. And we can find that as the number of diffusion steps increases, the baseline model equipped with prior-conditioning and prior-shifting mechanism improves more significantly on the Vina Score and SA.

**Effect of the Shifting Scales**    As mentioned in Equation (6), the shift $S_t$ at the time step $t$ consists of a coefficient $k_t$ and a 3-dim vector generated by a learnable neural network $\phi(\cdot)$, where the coefficient $k_t = \eta \cdot \sqrt{\bar{\alpha}} \cdot (1 - \sqrt{\bar{\alpha}})$ and $\eta$ is a hyper-parameter to adjust the scale of the shifts in the diffusion trajectory. To inverstigate the effect of different shifting scales, we set the $\eta$ to 5 values: (1) $\eta = 0$, (2) $\eta = 0.1$, (3) $\eta = 1$, (4) $\eta = 10$, (5) $\eta = \sqrt{\bar{\alpha}} \cdot (1 - \sqrt{\bar{\alpha}})$ and present the results in the Tab. 7. It worth noting that $\eta = 0$ indicates the prior-shifting mechanism is removed from IPDIFF. We

found that reducing the shifting scales in the whole diffusion trajectories ($\eta = 0.1$) can not bring gains to IPDIFF, and enlarging the shifting scales ($\eta = 10$) even hurts the performance of IPDIFF in property-related metrics. While highlighting the shifting scales in the middle diffusion trajectories ($\eta = \sqrt{\bar{\alpha}} \cdot (1 - \sqrt{\bar{\alpha}})$) will further improve the performance of IPDIFF on High Affinity. In practice, we set $\eta = 1$.

Table 7: The effect of the different shifting scales on binding-related metrics. ($\uparrow$) / ($\downarrow$) denotes a larger / smaller number is better. Top 2 results are highlighted with **bold text** and underlined text, respectively.

| Methods | Vina Score ($\downarrow$) | | Vina Min ($\downarrow$) | | Vina Dock ($\downarrow$) | | High Affinity ($\uparrow$) | |
|---|---|---|---|---|---|---|---|---|
| | Avg. | Med. | Avg. | Med. | Avg. | Med. | Avg. | Med. |
| baseline | -5.04 | -5.75 | -6.38 | -6.52 | -7.55 | -7.72 | 54.2% | 54.1% |
| $\eta = 0$ | -5.30 | -6.17 | -6.61 | -6.81 | -7.94 | -8.01 | 60.6% | 66.9% |
| $\eta = 0.1$ | -5.10 | -5.93 | -6.51 | -6.63 | -7.78 | -7.91 | 62.7% | 66.1% |
| $\eta = 1$ | **-6.42** | **-7.01** | **-7.45** | **-7.48** | **-8.57** | **-8.51** | 69.5% | **75.5%** |
| $\eta = 10$ | -5.84 | -6.41 | -6.87 | -6.90 | -7.91 | -8.07 | 60.9% | 61.1% |
| $\eta = \sqrt{\bar{\alpha}} \cdot (1 - \sqrt{\bar{\alpha}})$ | -6.03 | -6.71 | -7.26 | -7.31 | -8.48 | **-8.51** | **70.3%** | 74.0% |

**Reducing The Gap between Training and Sampling Processes**    As described in Secs. 4.2.1 and 4.2.2, the interaction prior utilized in the training and sampling processes of IPDIFF are obtained in different manners. Specifically, in the training process of IPDIFF, the ground-truth molecular protein-ligand pairs are fed into the pretrained IPNET to obtain the interaction prior, while in the sampling process of IPDIFF, the molecules predicted in the previous time step and given condition pockets are utilized to obtain the interaction prior. In order to reduce the gap between the interaction prior in training and sampling processes, especially when the molecules cannot be accurately estimated with large noise during the sampling process, we add Gaussian noises $\mathbf{z} \sim \mathcal{N}(0, \sigma^2\mathbf{I})$ to the position of each atom in molecules during the training of IPNET, and control the scale of noises through $\sigma$. The experimental results are present in Tab. 8. We observed that when no noises are introduced and the introduced noises too large or too small lead to limited improvements. In practice, we set the $\sigma = 0.5$.

Table 8: Effect of the noise scales in training process of IPNET. ($\uparrow$) / ($\downarrow$) denotes a larger / smaller number is better. Top 2 results are highlighted with **bold text** and underlined text, respectively.

| Methods | Vina Score ($\downarrow$) | | Vina Min ($\downarrow$) | | Vina Dock ($\downarrow$) | | High Affinity ($\uparrow$) | |
|---|---|---|---|---|---|---|---|---|
| | Avg. | Med. | Avg. | Med. | Avg. | Med. | Avg. | Med. |
| baseline | -5.04 | -5.75 | -6.38 | -6.52 | -7.55 | -7.72 | 54.2% | 54.1% |
| $\sigma = 0$ | -5.37 | -6.47 | -6.69 | -6.81 | -7.82 | -7.90 | 60.1% | 58.2% |
| $\sigma = 0.1$ | -5.63 | -6.80 | -7.10 | -7.36 | -8.17 | -8.35 | 64.7% | 67.0% |
| $\sigma = 0.5$ | **-6.42** | **-7.01** | **-7.45** | **-7.48** | **-8.57** | **-8.51** | **69.5%** | **75.5%** |
| $\sigma = 1$ | -3.89 | -6.05 | -6.20 | -6.75 | -7.84 | -7.97 | 60.0% | 58.4% |

**Performance of Generated Molecules with Different Number of Rotatable Bonds**    The Figure 6 presents the performance of generated molecules with the different number of rotatable bonds. And we can observed that our method can achieve superior performance compared with TargetDiff, even for the molecules with the large number of rotatable bonds ($> 10$),

**Time Complexity**    For investigating the sampling efficiency, we report the inference time of our model and other baselines for generating 100 valid molecules on average. Pocket2Mol, TargetDiff and DecompDiff use 2037s, 1987s and 3218s, and our IPDIFF takes 3063s.

# F    FUTURE WORK

In this paper, we introduce a paradigm that combines a pretrained protein-ligand interaction model with the diffusion models for SBDD. In Tab. 3, we have present that our IPNET can effectively serve the downstream diffusion model even when it is solely pre-trained on sequential data. This is

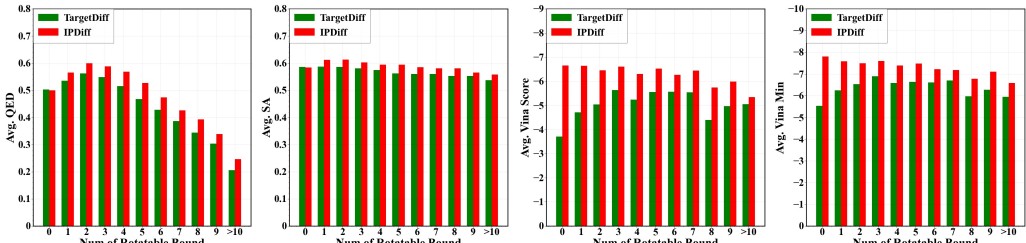

Figure 6: The performance of generated molecules with different number of rotatable bonds.

significant because precise geometric data is often scarce, while sequence data is abundant. This finding suggests that we can further explore the integration of self-supervised or weakly-supervised learning techniques in the training of IPNET solely relying on abundant sequential data, for extracting general interaction-related information and effectively serving the downstream diffusion model. These aspects make our model scalable and open up new possibilities for leveraging large-scale sequence data in SBDD tasks. Moreover, It is necessary to evaluate our methods from different evaluation perspectives. We will provide comprehensive experiments and analysis on the other metrics, such as PoseCheck (Harris et al., 2023) and PoseBusters (Buttenschoen et al., 2023).

## G  MORE RESULTS

We provide the visualization of more ligand molecules generated by IPDIFF, comparing to both reference and TargetDiff (Guan et al., 2023a), as shown in Figure 7.

We provide the source files containing generated molecules and the evaluation code that can reproduce the results in Tab. 2 in the supplementary material.

We are committed to open source the code of training and inference as well as the pretrained model upon paper acceptance.

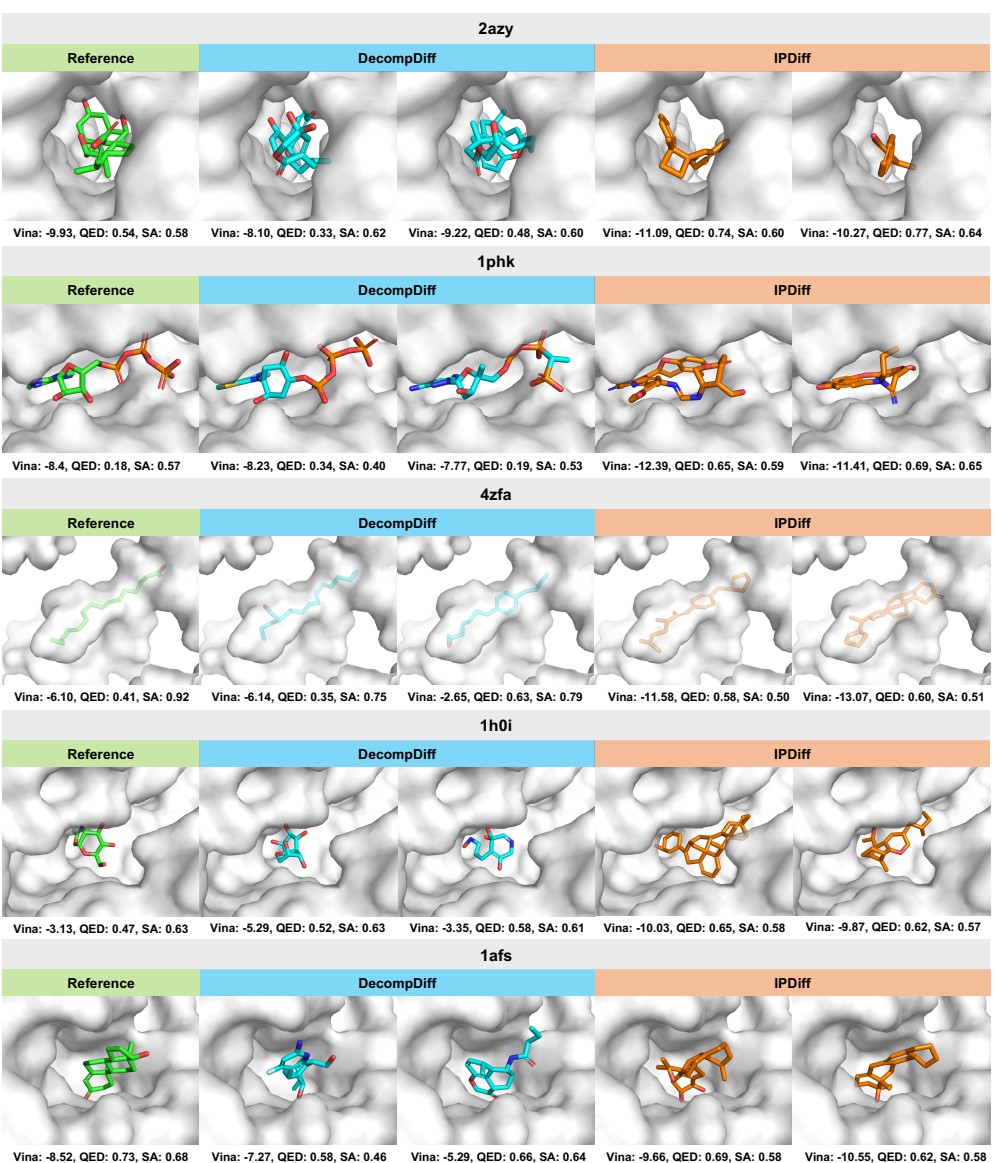

Figure 7: More examples of generated ligands for protein pockets. Carbon atoms in reference ligands, ligands generated by DecompDiff (Guan et al., 2023b) and IPDIFF are visualized in green, blue and orange, respectively. We report Vina Score, QED and SA for each molecule.

