# OpenReview forum: "Protein-Ligand Interaction Prior for Binding-aware 3D Molecule Diffusion Models"
_ICLR.cc/2024/Conference — ICLR 2024 poster_

### Official Review · Reviewer_F898 · 2023-10-21

**Soundness:** 3 good
**Presentation:** 3 good
**Contribution:** 3 good
**Rating:** 6
**Confidence:** 3

**Summary:**

This paper introduces a novel approach to 3D molecule generation for structure-based drug design using diffusion models. Recognizing the inconsistency in existing diffusion models that incorporate protein-ligand interaction information predominantly in the reverse process, the authors propose the Interaction Prior-guided Diffusion model (IPDiff). IPDiff integrates geometric protein-ligand interactions into both the diffusion and sampling processes. The methodology involves pretraining an interaction prior network (IPNet) using binding affinity signals, followed by leveraging this network to adjust molecule diffusion trajectories (termed "prior-shifting") and enhance the binding-aware molecule sampling process (referred to as "prior-conditioning"). Empirical studies on the CrossDocked2020 dataset demonstrate that IPDiff can effectively generate molecules with realistic 3D structures that exhibit superior binding affinities towards target proteins, while preserving essential molecular properties.

**Strengths:**

1. The concept of "Prior-Shifting" is innovative. Adjusting the forward process based on specific data can indeed enhance the efficiency of reverse diffusion.

2. The paper's presentation is well-structured and coherent.

**Weaknesses:**

1. The author's evaluation seems limited to the CrossDocked2020 benchmark. For a more comprehensive assessment, it would be beneficial to test the method across multiple benchmarks.

2. “In the forward process, the ways of injecting noises are the same for all training samples with different target proteins.” While the statement mentions that the noise injection methods are consistent for all training samples with varying target proteins, it's essential to recognize that the forward diffusion process might differ. Even if the noise injection methods remain the same, the original samples' distinctiveness ensures varied forward diffusion outcomes.

3. It's noteworthy that there have been prior studies that learned the beta parameters for forward diffusion. A comparative analysis with those works would add depth to this study.

4. Based on"Analog bits: Generating discrete data using diffusion models with self-conditioning," the novelty of the prior conditioning contribution in this paper appears less significant.

5. I am also quite curious whether this prior-shifting idea can be used in class-conditional image generation as different classes are very different.

**Questions:**

See Weakness.

---

> ### Author Response · Authors · 2023-11-17
> **Response to Reviewer F898**
>
> *We sincerely thank you for your time and efforts in reviewing our paper, and your valuable feekback. We are happy to see that you recognize the concept of prior-shifting proposed in our paper is innovative, and the paper's presentation is well-structured and coherent. Please see below for our responses to your comments, and the revised texts are marked in red.*
>
> **Q1: It would be beneficial to test the method across multiple benchmarks.**
>
> A1: We added a comparison to BindingMOAD dataset which is used in DiffSBDD [1], and the other SBDD methods only conducted comparisons on the CrossDocked2020. We preprocessed the data following DiffSBDD. The results are shown in the following table. The results demonstrate that our IPDiff can achieve the best performance by comparing with DiffSBDD and TargetDiff [2].
>
> | Methods    | Vina Score (-) | QED (+)  | SA (+)   | Diversity (+) |
> | ---------- | -------------- | -------- | -------- | ------------- |
> | DiffSBDD   | -5.20          | 0.47     | 0.57     | **0.91**      |
> | TargetDiff | -5.49          | 0.46     | 0.56     | 0.89          |
> | IPDiff     | **-6.11**      | **0.50** | **0.61** | 0.88          |
>
> **Q2: “In the forward process, the ways of injecting noises are the same for all training samples with different target proteins.” Even if the noise injection methods remain the same, the original samples' distinctiveness ensures varied forward diffusion outcomes.**
>
> A2: We want to convey that in previous diffusion models, the noise added to the atom of ligand is sampled from a normal distribution that is independent of the given pocket. In contrast, our method takes into account the interaction information between the protein and ligand during the noise addition process.
>
> **Q3: A comparative analysis with works that learned the beta parameters for forward diffusion would add depth to this study.**
>
> A3: We conduct a experiment that makes the beta parameter in TargetDiff learnable. The performance comparison between TargetDiff, TargetDiff with learnable beta and our IPDiff are present below. The results demonstrate that simply make beta parameters learnable for the forward diffusion can not improve the performance of diffusion models for SBDD.
>
>
> | Methods                     | Vina Score (-) | QED (+)  | SA (+) |
> | --------------------------- | -------------- | -------- | ------ |
> | TargetDiff                  | -5.47          | 0.48     | 0.58   |
> | TargetDiff + learnable beta | -5.33          | 0.49     | 0.58   |
> | IPDiff                      | **-6.42**      | **0.52** | **0.61**   |
>
> **Q4: Compared with self-conditioning proposed in Analog, the novelty of the prior conditioning contribution in this paper appears less significant.**
>
> A4: The novel technical contributions of our proposed prior-conditioning mechanism are: (1) It incorporates protein-ligand interaction information modeled in the pretrained prior network into the molecular diffusion model while self-condition can only consider self-prediction molecular information and is unable to model the critical protein-molecule geometric interactions.
> (2) It introduces a new paradigm that combines the protein-ligand pretraining with protein-conditioned molucular diffusion model, fundamentally different from previous diffusion-only methods.
> And we have conducted a quantitative comparison between our prior-conditioning and self-conditioning. The results in **Table 4** of the manuscript demonstrate that self-conditioning is not effective in SBDD tasks since it can not provide the interaction information between ligands and proteins.
>
> **Q5: I am also quite curious whether this prior-shifting idea can be used in class-conditional image generation as different classes are very different.**
>
> A5: From a broader perspective, IPDiff represents a cross-modal conditional diffusion model, which is designed to emphasize the interaction between different modalities in the diffusion process and utilize it as a condition. We conduct a preliminary experiment on CIFAR-10 where we incorporate class-image semantic interactions (we obtain the shifts from the combination of class embeddings and image embeddings) into the image diffusion process, the results are in the table below. We can find that in class-conditional image generation, our model can still improve the baseline diffusion model, demonstrating the satisfying generalization ability of our alrorithm.
>
> | Methods                               | FID $\downarrow$ |
> | ------------------------------------- | -------- |
> | DDPM                                  | 3.17     |
> | DDPM + class-image correlation (Ours) | **2.95** |
>
>
> **Reference**:
>
> [1] Structure-based Drug Design with Equivariant Diffusion Models, arXiv:2210.13695, 2022.
>
> [2] 3d Equivariant Diffusion for Target-aware Molecule Generation and Affinity Prediction, ICLR, 2023a.

---

> ### Author Response · Authors · 2023-11-20
> **Gentle Reminder**
>
> Dear Reviewer F898,
>
> We sincerely appreciate the time and effort you have dedicated to reviewing our paper. Your insightful questions and valuable feedback have been extremely beneficial.
>
> **In response, we have (1) provided the performance comparison on another benchmark (2) provided the performance comparisons by extending our methods to other application (i.e., class-to-image generation) for illustrating a boarder impact in AI community.**
>
> As the discussion period is approaching its conclusion in two days, we kindly request, if possible, that you review our rebuttal at your earliest convenience. Should there be any further points that require clarification or improvement, please know that we are fully committed to address them promptly.
>
> Thank you once again for your invaluable contribution to our research.
>
> Warm Regards,
>
> The Authors

---

> > ### Comment · Reviewer_F898 · 2023-11-21
> > **Thanks for your feedback.**
> >
> > Thanks for the reviewer's feedback. It did clarify some of my concerns. I tend to keep my original score for acceptance.

---

### Official Review · Reviewer_AJPX · 2023-10-30

**Soundness:** 2 fair
**Presentation:** 3 good
**Contribution:** 3 good
**Rating:** 5
**Confidence:** 3

**Summary:**

In this paper, the authors propose to augment standard diffusion-based models for pocked-conditioned 3D ligand generation. The proposed approach, called IPDiff, leverages protein-ligand interactions in both the forward and reverse diffusion processes. The protein-ligand interaction is learned with an auxiliary network (called IPNet) and trained separately on a PDBBind. The authors show good results (on standard metrics) on CrossDocked2020 dataset.

**Strengths:**

- The main idea of the paper—to leverage protein-ligand interactions to improve generation of 3D ligands—is novel and well motivated.
- The paper shows that ligand generation conditioned on pocket can be improved (at least in the case of diffusion models) if one can successfully leverage protein-ligand interactions.
- The proposed approach achieves good results on benchmark metrics/datasets (however, this method requires labeled data and one extra trained network, so comparisons aren’t really apples-to-apples)

**Weaknesses:**

- The proposed method improves over the baseline (diffusion model conditioned on pocket) by relying on supervised learning on PDBbind. This makes the model more complicated to train/use and, more importantly, more difficult to scale. These points needs to be mentioned in the main paper.
- IPNet is trained on PDBBind and then it is used as a feature extractor to the training of the diffusion model. What if PDBBind contains samples that are similar (or identical) to those on the test set of CrossDocked? If this is the case, the good performance could be due to some kind of “leaking” of information from the test set to the training set. The authors did not mention anything about this on the manuscript.
- The authors use the same metrics reported on previous work. Although this is good for benchmarking, it is well now by the community that these metrics are not great. For example, recent work (eg, PoseCheck and PoseBusters) propose other metrics that can better assess quality of generated molecules. It would be very nice the authors would have shown results on these metrics.
- Since the authors leverage protein-ligand interactions, it would be nice to compare how good the generated molecules are in terms of protein-ligand interactions with the pockets. Posecheck (mentioned above) proposes a metric to measure to compute this, but many other could be imagined.

**Questions:**

- Please see the comments on "Weaknesses" above.
- What is the objective used to train IPNet? Was it mean square errors? What is the ground-truth label and what is its range? I feel some details about the training of IPNet are missing.
- In the last paragraph of the Introduction, the authors say that “IPDiff is theoretically able to achieve better likelihood compared to previous diffusion models”. Can the authors elaborate on this?
- Between equation 10 and 11, the authors mention that they need to approximate X_0^M and V_0^M (since we dont have access to ground truth during generation). How bad is this approximation? How much performance is lost because of this? Would it be possible to do some experiment to analyse it?
- Based on the results on the tables of this paper, the proposed method has better metrics than the reference. This seems a bit strange to me and points to the fact that the metrics used are probably not very informative/useful. What does this mean in the opinion of the authors?

---

> ### Author Response · Authors · 2023-11-17
> **Response to Reviewer AJPX (Part 1/2)**
>
> *We sincerely thank you for your time and efforts in reviewing our paper, and your valuable feekback. We are happy to see that you **recognize the idea proposed in our paper is novel and well-motivated, the performance improvements are promising**. Please see below for our responses to your comments, and **the revised texts are denoted in red**.*
>
> **Q1: however, this method requires labeled data and one extra trained network, so comparisons aren’t really apples-to-apples.**
>
> A1: It is important to note that one of our baselines, DecompDiff, similarly incorporates external knowledge by using the extra software AlphaSpace to extract ligand-agnostic decomposed priors from the given pocket for the subsequent molecular generation process, while our IPNet ustilizes learnable interaction network to automatically model the protein-ligand binding prior, which is more flexible to use.
> Besides, compared to the DecompDiff that relies on the labeled geometric data, our IPNet has the potential to scale, as the interaction clues can be also learned from the sequential data (please refer to **Table 3** in the manuscript), which is easier to acquire.
> Moreover, the decomposed priors in DecompDiff are solely related to proteins, ignoring the fact that different molecules may suggest different protein-molecule interactions. In contrast, our method fully considers the interaction between given proteins and molecules as conditions during the generation process, which is more relevant to the task at hand.
>
> **Q2: The proposed method improves over the baseline (diffusion model conditioned on pocket) by relying on supervised learning on PDBbind. This makes the model more complicated to train/use and, more importantly, more difficult to scale**
>
> A2: Indeed, we utilize the supervision of binding affinity in PDBBind during training. However, we do not think that this will make our model difficult to use or train. It is important to note that the binding affinity is only used as a supervision in the training of IPNet, and the diffusion model in the second stage does not rely on binding affinity. The purpose of IPNet is to mine some general interaction clues, and we can arbitrarily design the diffusion model without changing the IPNet. **Additionally, our method introduces a paradigm that combines a pretrained protein-ligand interaction model with diffusion models for SBDD, which has a boarder impact on AI-for-science community.** We have conducted experiments to demonstrate that our IPNet can effectively serve the downstream diffusion model even when it is solely pre-trained on sequential data (please refer to **Table 3** in the manuscript). This is significant because precise geometric data is often scarce, while sequential data is abundant. This finding suggests that we can further explore the integration of Self-Supervised Learning or Weakly-Supervised Learning techniques in the training of IPNet solely relying on abundant sequential data, for extracting general interaction-related information and effectively serving the downstream diffusion model. These aspects make our model scalable and open up new possibilities for leveraging large-scale sequential data in SBDD. We are grateful for your insightful suggestion, and we thoroughly explained and discussed it in the **Appendix-F** revised manuscripted as part of our future work.
>
> **Q3: What if PDBBind contains samples that are similar (or identical) to those on the test set of CrossDocked?**
>
> A3: We have **excluded the molecules used in testing and evaluation of IPDiff from the PDBBind v2016 dataset for IPNet** according to the calculated Tanimoto similarity between two molecules. After that, the average Tanomoto similarity between these datasets is only **0.12\%**. Additionally, we have also calculated the Tanimoto similarity between the molecules generated by IPDiff and the molecules in the training of IPNet, and the average Tanimoto similarity is only **0.079\%**. Based on these, we believe that there is no issue of information leakage in our method.
>
> **Q4: It would be nice to compare how good the generated molecules are in terms of protein-ligand interactions with the pockets. Posecheck (mentioned above) proposes a metric to measure to compute this.**
>
> A4:
> We conduct a comparison between our IPDiff and TargetDiff on the PoseCheck as present below. We can observe that our method achieves the superior performance compared with TargetDiff. We have cited PoseCheck [4] in the **Appendix-F** of the revised manuscript (marked in red), and we will provide more comprehensive evaluation and analysis of our method on PoseCheck for the future work.
>
> |  Methods   | Clashes (-) | Strain Energy (-) | No. Acceptors (+) | No. Donors (+) |
> | ---------- | ----------- | ----------------- | ----------------- | -------------- |
> | TargetDiff | 10.7        | 1404.8            | 0.87              | 0.33           |
> | IPDiff     | **8.1**     | **1105.9**        | **0.95**          | **0.42**       |

---

> ### Author Response · Authors · 2023-11-17
> **Response to Reviewer AJPX (Part 2/2)**
>
> **Q5: I feel some details about the training of IPNet are missing.**
>
> A5: We have reported more details about the training of IPNet in the **Appendix-D.1 of the updated manuscript marked in red**, please refer to it.
>
> **Q6: “IPDiff is theoretically able to achieve better likelihood compared to previous diffusion models”. Can the authors elaborate on this?**
>
> A6: We show IPDiff is theoretically able to achieve better likelihood than previous diffusion models.  As the exact likelihood is intractable, we aim to compare the optimal variational bounds for negative log likelihoods (NLL). The objective function of IPDiff at time step t is $E_{q_{\theta_2}}D_{KL}(q_{\theta_2}(x_{t-1}|x_t,x_0,F_0, P)||p_{\theta_1}(x_{t-1}|x_t,F_{0|t+1},F_{0|t},P))$, and its optimal solution is
> $$\begin{aligned}&\min_{\theta_1,\theta_2}E_{q_{\theta_2}}D_{KL}(q_{\theta_2}(x_{t-1}|x_t,x_0,F_0,P)||p_{\theta_1}(x_{t-1}|x_t,F_{0|t+1},F_{0|t},P)) \\\\
> &=\min_{\theta_2}[\min_{\theta_1}E_{q_{\theta_2}}D_{KL}(q_{\theta_2}(x_{t-1}|x_t,x_0,F_0,P)||p_{\theta_1}(x_{t-1}|x_t,F_{0|t+1},F_{0|t},P)) \\\\
> &\leq\min_{\theta_1}E_{q}D_{KL}(q(x_{t-1}|x_t,x_0,P)||p_{\theta_1}(x_{t-1}|x_t,P)),\end{aligned}$$
> where $\min_{\theta_1}E_{q}D_{KL}(q(x_{t-1}|x_t,x_0,P)||p_{\theta_1}(x_{t-1}|x_t,P))$ is the optimal loss of previous diffusion models that do not use interactive representations $F$ in the forward process. Similar inequality can be obtained for t=1:
> $$\begin{aligned}
>     &\min_{\theta_1,\theta_2} E_{q_{\theta_2}}-\log p_{\theta_1}(x_0|x_1,F_{0|1},P)\\\\
>     &\leq \min_{\theta_1}E_{q}-\log p_{\theta_1}(x_0|x_1,P)
> \end{aligned}$$
> As a result,  we have the following inequality by summing up the objectives at all time steps:
> $$\begin{aligned} &-E_{q(x_0|P)}\log p_{\theta_1}(x_0|P) \\\\
> &\leq \min_{\theta_1,\theta_2} \sum_{t>1} E_{q_{\theta_2}} D_{KL}(q_{\theta_2}(x_{t-1}|x_t,x_0,F_0,P)||p_{\theta_1}(x_{t-1}|x_t,F_{0|t+1},F_{0|t},P))+E_{q_{\theta2}}-\log p_{\theta_1}(x_0|x_1,F_{0|1},P)+C\\\\
> &\leq \min_{\theta_1} \sum_{t>1} E_{q}D_{KL}(q(x_{t-1}|x_t,x_0,P)||p_{\theta_1}(x_{t-1}|x_t,P))+E_{q}-\log p_{\theta_1}(x_0|x_1,P)+C\end{aligned}$$
> where C is a constant defined by $\sqrt{\bar{\alpha}_T}$. Hence, IPDiff has a tighter bound for the NLL, and thus theoretically capable of achieving better likelihood, compared with the previous diffusion models.
>
> **Q7: Between equation 10 and 11, the authors mention that they need to approximate $X_0^M$ and $V_0^M$. How bad is this approximation? How much performance is lost because of this? Would it be possible to do some experiment to analyse it?**
>
> A7: Approximating $X_0^M$ and $V_0^M$ is a necessary step in the original diffusion model, and the closer the time step t is to T, **the closer the $X_0^M$ approximated by the model at time step t is to the noise**. In order to further assess how much performance lost in our IPDiff brought by the inaccurate approximation and interaction prior, we conducted two experiments: (1) Exp1: we use reference ligands to extract the interaction prior in the first 200 time steps of the sampling process, and the remaining steps were consistent with the original sampling process in IPDiff; (2) Exp2: we use randomly generated atomic positions and types to extract interaction prior for IPDiff in the first time 200 steps of the time step. The performance are presented below. We can observe that the vina score of Exp1 is slightly improved, but the diversity is simultaneously decreased. And the performance of Exp2 is largely degraded.
> In conlusion, although the interaction prior introduced by the approximated $X_0^M$ and $V_0^M$ is not optimal, such approximation can still bring effective interaction clues for facilitating protein-based molecular generation.
>
> | Methods | Vina Score (-) | Diversity (+) |
> | ------- | -------------- | ------------- |
> | IPDiff  | -6.42          | **0.74**      |
> | Exp1    | **-6.45**      | 0.71          |
> | Exp2    | -4.30          | **0.74**      |
>
>
> **Q8: The proposed method has better metrics than the reference. This seems a bit strange to me and points to the fact that the metrics used are probably not very informative/useful. What does this mean in the opinion of the authors?**
>
> A8: The reference ligands in CrossDocked are obtained by recombining protein-ligand pairs from PDBBind and then redocking them. However, some of these reference ligands do not bind perfectly with proteins, indicating that their performance does not represent the upper bound. And the similar comparsion results between the generated molecules with the references are also presented in previous methods [1-3].
>
> [1] E(n) Equivariant Graph Neural Networks, ICML, 2021.
>
> [2] 3d Equivariant Diffusion for Target-aware Molecule Generation and Affinity Prediction, ICLR, 2023a.
>
> [3] DecompDiff: Diffusion Models with Decomposed Priors for Structure-Based Drug Design, ICML, 2023.
>
> [4] PoseCheck: Generative Models for 3D Structure-based Drug Design Produce Unrealistic Poses, NeurIPS Workshop, 2023.

---

> ### Author Response · Authors · 2023-11-20
> **Gentle Reminder**
>
> Dear Reviewer AJPX,
>
> We sincerely appreciate the time and effort you have dedicated to reviewing our paper. Your insightful questions and valuable feedback have been extremely beneficial.
>
> **In response, we have (1) provided the theoretical proofs that our IPDiff has capability to achieve better likelihood compared to previous diffusion models, (2) provided more experiments and analysis of prior-shifting mechanism, (3) clarified the scalability of IPDiff, (4) evaluated the performance with more evaluation metrics to address your queries and concerns.**
>
> As the discussion period is approaching its conclusion in two days, we kindly request, if possible, that you review our rebuttal at your earliest convenience. Should there be any further points that require clarification or improvement, please know that we are fully committed to address them promptly.
>
> Thank you once again for your invaluable contribution to our research.
>
> Warm Regards,
>
> The Authors

---

> ### Comment · Reviewer_AJPX · 2023-11-22
> **Answer to authors' rebuttal**
>
> We thank the authors for the answers to questions above.
>
> - **Apples-to-apples comparisons:** we agree that both DecompDiff and IPDiff are extensions (by leveraging more information) of point cloud-based diffusion models (eg TargetDiff and SBDDDiff).
> - **IPNet:** thank you for the clarifications about the architecture and other questions. This network is a key component of the model and it is good to see more details about it on the paper.
> - **PDBBind/CrossDocked leaked information:** Thank you for the clarification. This is a very important issue for model benchmarking and it needs to be explicitly mentioned in the text. I did not see any comment about this on the manuscript. It is also important to know how much the PDBBind dataset is reduced after the preprocessing.
> - **Metrics:**
>   - Where are the DecompDiff results? As we agreed above, comparing with TargetDiff is not very  meaningful (since IPDiff is an augmentation of TargetDiff in ways)
>   - The table shown by the authors is not very informative and does not allows us to have many conclusions. Are those numbers mean? median? They usually have large variance and are better shown with plots (as in the paper). How they compare to the other baselines (particularly the "ground truth")?
>   - The main contribution of the paper is leveraging PL-interactions, yet the authors only show 2 out of 4 of the proposed interactions metrics.
>   - Moreover, both clash and strain energy are still very bad compared to other methods (besides TargetDiff).
> - **Comparison with docking baseline:** The authors argue that their approach ids better than the baseline based the results on Table 1. However, even with the incomplete results in Posecheck metrics, we can see that the generated molecules have much more clashes and strain energy than the simple baseline. This indicates that the proposed method is not as good as it sounds in practice.
>
> Based on the comments above, I will keep my rating.

---

> > ### Author Response · Authors · 2023-11-23
> > **Response to Reviewer AJPX**
> >
> > Dear Reviewer AJPX,
> >
> > Thanks for your further comments, we carefully response to your comments as follows.
> >
> > **Q1: I did not see any comment about this on the manuscript is also important to know how much the PDBBind dataset is reduced after the preprocessing.**
> >
> > A1: There are originally 3767 complexes in the training set of PDBBindv2016. In our experiments, we filter molecules in this training set by Tanimoto similarity with the molecules in the testing set of CrossDocked2020 to avoid information leakage, and the similarity threshold was set to 0.1. As a result, there are 23 complexes filtered out from the training set which exceed the similarity threshold. We have added this detailed process in the revised Appendix-D.1 and marked in red. Please kindly check this out.
> >
> > **Q2: Where are the DecompDiff results? Are those numbers mean? median?  How they compare to the other baselines (particularly the "ground truth")?**
> >
> > A2: We report the mean value of Clashes and Interactions (No. Acceptor, No. Donors, vdWs and Hydrophobic interactions), and the median value for Strain Energy due to its large ranges as mentioned in PoseCheck [8]. The complete comparisons between IPDiff, TargetDiff, DecompDiff, and reference are presented as follows:
> >
> > For the generated conformations of ligands:
> >
> > | PoseCheck  | Clashes (-) | Strain Energy  (-) | No. Acceptor (+) | No. Donors (+) | vdWs (+) | Hydrophobic interactions (+) |
> > | ---------- | :----: | :----: | :----: | :----: | :----: | :----: |
> > | TargetDiff | 10.7| 1404.8 | 0.87 | 0.33 | 7.7| 5.5 |
> > | DecompDiff | 8.3 | 1204.1| 0.83  | 0.36| 7.2| 4.2|
> > | IPDiff     | **8.1**| **1105.9** | **0.95**| **0.42** | **8.0**  | **5.8**|
> >
> > For the redocked comformations of ligands:
> >
> > | PoseCheck  | Clashes (-) | No. Acceptor (+) | No. Donors (+) | vdWs (+) | Hydrophobic interactions (+) |
> > | ---------- | :----:| :----: | :----: | :----: | :----: |
> > | Reference  | 5.3         | **2.12**         | **1.31**       | 6.7      | 4.7  |
> > | TargetDiff | 4.8         | 0.96             | 0.44           | 6.9      | 5.3 |
> > | DecompDiff | 3.7         | 0.71             | 0.28           | 5.3      | 4.8  |
> > | IPDiff     | **3.4**     | 0.99             | 0.48           | **7.4**  | **5.7** |
> >
> >
> > Our method outperforms its counterparts, TargetDiff and DecompDiff, in almost all metrics, which has demonstrated the effectiveness of our method. Notably, the evaluation metrics used in our paper and in the rebuttal assess the quality of the generated molecules from different aspects, and reveal the consistent superiority of our method over previous works.
> >
> > **Q3: The main contribution of the paper is leveraging PL-interactions, yet the authors only show 2 out of 4 of the proposed interaction metrics. Moreover, both clash and strain energy are still very bad compared to other methods (besides TargetDiff). More results about the comparison with docking baseline.**
> >
> > A3: Following your suggestions, we compare with previous methods regarding all the proposed interaction metrics in PoseCheck and report the results **in the table above**. From the results, we can find that IPDiff is superior to all previous diffusion-based SBDD methods. We attribute this to our pre-trained protein-ligand interaction prior and prior-conditioning/shifting mechanisms for diffusion generation.
> >
> > We would like to emphasize again that we **strictly follow the experimental settings and evaluation metrics of previous methods [1-7] to ensure fair comparisons**. Our method mainly focuses on improving the binding affinities towards the protein targets,  **IPDiff surpasses all previous diffusion-based and non-diffusion methods** in binding-related metrics. Based on the results in both the paper and rebuttal, we believe IPDiff has convincingly demonstrated its efficacy as a new diffusion-based SBDD method.
> >
> >
> > **References**
> >
> > [1] Peng et al., Pocket2mol: Efficient Molecular Sampling based on 3d Protein Pockets, ICML, 2022.
> >
> > [2] Schneuing et al., Structure-based Drug Design with Equivariant Diffusion Models, arXiv:2210.13695, 2022.
> >
> > [3] Guan et al., 3d Equivariant Diffusion for Target-aware Molecule Generation and Affinity Prediction, ICLR, 2023.
> >
> > [4] Guan et al., DecompDiff: Diffusion Models with Decomposed Priors for Structure-Based Drug Design, ICML, 2023.
> >
> > [5] Luo et al., A 3d Generative Model for Structure-based Drug Design, NeurIPS, 2021.
> >
> > [6] Ragoza et al., Generating 3D molecules Conditional on Receptor Binding Sites with Deep Generative Models, Chem Sci, doi: 10.1039/D1SC05976A.
> >
> > [7] Liu et al., Generating 3d Molecules for Target Protein Binding, ICML, 2022.
> >
> > [8] Harris et, al., PoseCheck: Generative Models for 3D Structure-based Drug Design Produce Unrealistic Poses, NeurIPS Workshop, 2023.
> >
> > ----
> >
> > Should there be any further points that require clarification or improvement, please know that we are fully committed to addressing them promptly.
> >
> > Thank you once again for your invaluable contribution to our research.
> >
> > Warm Regards,
> >
> > The Authors

---

### Official Review · Reviewer_qHa3 · 2023-11-03

**Soundness:** 3 good
**Presentation:** 3 good
**Contribution:** 3 good
**Rating:** 8
**Confidence:** 4

**Summary:**

This work is focused on improving the quality of the generated molecular pose within the protein pocket by3D-diffusion protein-ligand models. Authors argue that the different utilisation of the protein-ligand interactions by the forward and reverse process impacts the quality of the  generated structures.  In particular this is because the differences of the pocket binding sites in training samples are neglected by the forward process while leveraged during the reverse process. Authors explored this problem by proposing a solution where they adapt the trajectories such that these contain information about the protein-ligand interactions into the forward process by altering the drift of the diffusion trajectory based on the interactions while conditioning the reverse diffusion trajectory on the estimates of the protein-ligand interaction.

Authors introduce atom-wise cross attention layer to model intra- and intermolecular interactions of protein ligand pairs which they use  foir pre-training and use the pretrained representations in the diffusion model.
Authors further  provide comparisons with various methods to demonstrate the strengths of their proposed approach.

**Strengths:**

- The paper is very clearly written, the authors identified an interesting problem for which they proposed a solution and demonstrated by well designed experiments, that their proposed solution works well.
- The derivations of actions on the forward and backward diffusion kernels  and training objectives are provided in the Appendix, very clearly written and easy to follow.
- The ablation study exploring the role of the scaller within the drift correction  of the forward process is provided.
- The experiments are sufficiently described and most likely reproducible.

**Weaknesses:**

- There are minor typos on various places of the manustript.
- The edge cases such as molecules with high number of rotable bonds are not investigated.

**Questions:**

1. What is the difference in the chemical diversity of the datasets used for training IPNET and IPDfiff? e.g. different scaffolds, molecules with large flexibility, etc. Were the poses used in testing and evaluation of  IPDiff excluded from the training set for IPNet?

2. How does the model perform when generating the molecules with larger numbers of rotatable bonds? How does the model generalises into unseen targets and novel pockets?

3. What is the percentage of the  novel molecules generated by the IPDiff?

4. IPDiff generates molecules in average 31s per molecule. How fast is the docking algorithm on the same hardware and similar sized structures?

---

> ### Author Response · Authors · 2023-11-17
> **Response to Reviewer qHa3**
>
> *We sincerely thank you for your time and efforts in reviewing our paper, and your valuable feekback. We are happy to see that you **recognize that the proposed IPDiff novelty and effectiveness, theoretical proofs in our paper are clearly written and easy to follow, experiments are sufficient and reproducible**. Please see below for our responses to your comments, and **the revised texts are denoted in red**.*
>
> **Q1: What is the difference in the chemical diversity of the datasets used for training IPNET and IPDfiff? Were the poses used in testing and evaluation of IPDiff excluded from the training set for IPNet?**
>
> A1: CrossDocked2020 (the dataset used by IPDiff) is an expanded dataset obtained by recombining protein-ligand pairs from PDBBind (the dataset used by IPNet) and then redocking them. **Therefore, the chemical diversity of these two datasets is consistent.
> And we excluded the molecules used in testing and evaluation of IPDiff from the PDBBind v2016 dataset for IPNet** according to the calculated Tanimoto similarity between two molecules. After that, the average Tanomoto similarity between these datasets is only 0.12\%. Additionally, we have also calculated the Tanimoto similarity between the molecules generated by IPDiff and the molecules in the training of IPNet, and the average Tanimoto similarity is only **0.079\%**. Based on these, we believe that there is no issue of information leakage in our method.
>
> **Q2: How does the model perform when generating the molecules with larger numbers of rotatable bonds? How does the model generalises into unseen targets and novel pockets?**
>
> A2: (1) We report the performance in generating molecules with different numbers of rotatable bonds in the **revised Appendix-E and marked in red**, please refer to it.
> (2) All data preprocess and experimental settings in our paper follows previous methods. Specifically, the CrossDocked2020 dataset is partitioned into non-overlapping training and testing sets. The performance on the testing set of CrossDocked2020 can reflect the ability of our model in generalizing to unseen targets and novel pockets.
>
> **Q3: What is the percentage of novel molecules generated by the IPDiff?**
>
> A3: The percentage of novel molecules generated by the IPDiff is 99.999971\%.
>
> **Q4: How fast is the docking algorithm on the same hardware and similar sized structures?**
>
> A4: The average time consumed for docking a molecule to the given pocket using the Vina Dock algorithm on the same hardware is 27.9 seconds.

---

> ### Author Response · Authors · 2023-11-20
> **Gentle Reminder**
>
> Dear Reviewer qHa3,
>
> We sincerely appreciate the time and effort you have dedicated to reviewing our paper. Your insightful questions and valuable feedback have been extremely beneficial.
>
> **In response, we have (1) investigated the performance of molecules with a high number of rotatable bonds, (2) clarified the experimental settings and the characteristics of the datasets to address your queries and concerns.**
>
> As the discussion period is approaching its conclusion in two days, we kindly request, if possible, that you review our rebuttal at your earliest convenience. Should there be any further points that require clarification or improvement, please know that we are fully committed to address them promptly.
>
> Thank you once again for your invaluable contribution to our research.
>
> Warm Regards,
>
> The Authors

---

> > ### Comment · Reviewer_qHa3 · 2023-11-21
> >
> > Dear authors,
> >
> > I am happy with the responses which you provided to my questions and to the questions of the fellow reviewers. I am going to keep my acceptance score unchanged.

---

### Official Review · Reviewer_ssPe · 2023-11-05

**Soundness:** 2 fair
**Presentation:** 2 fair
**Contribution:** 2 fair
**Rating:** 6
**Confidence:** 3

**Summary:**

The authors present IPDiff for structure based drug design - wherein they propose to factor receptor-ligand pocket interactions into the noising and de-noising of the diffusion models. They do so by training IPNet which models the receptor and ligand as graphs - and is trained to predict the binding affinity. The trained IPNet is then used towards what the authors term as prior shifting - where the diffusion trajectories of forward process are influenced by the interactions. The authors then design a prior-conditioning step to enhance the reverse process by conditioning the denoising of ligand molecules on the previously estimated protein-ligand interactions.

**Strengths:**

1. The ideas to perform prior shifting and prior-conditioning for structure based drug design are novel to the best of my knowledge.
2. The proposed mechanism achieves SOTA performance on CrossDocked2020 benchmark, and is also able to generate
the molecules with -6.42 Avg. Vina Score (in comparison to -5.67 from prior baselines) while maintaining proper molecular properties. The authors also present the impact of prior shifting and conditioning in their model analysis section.

**Weaknesses:**

The clarity of the paper could be improved. Specifically
1. The paper cites Sattoras et al - and says its an SE(3) Equivariant neural network - whereas it is E(N) GNN. This is important to consider in the context of protein molecules as chirality is an important property of protein molecules.
2. The paper explicitly uses message passing only over the neighboring nodes as a part of IPNet. This is in contrast to E(N) GNN wherein, messages are passed between all pairs of nodes. Again this is crucial is this what ensures it is a rigid body and allows for E(3) equivariance/ invariances. The authors haven't show their version is SE(3) equivariant.
3. In section 4.2.1 it is unclear how it S is trained to ensure the molecule is not distorted to impossible molecules - as its not a simple translation matrix but positions for every atom - and can lead to arbitrary invalid deformations.
4. Using the pocket structure for the diffusion process is counter intuitive. When we start with the receptor for an unknown ligand (small molecules which we wish to discover) - the pocket or the holo structure of the receptor is not a single conformation (but an ensemble). Moreover, without the presence of the ligand - we only have access to the apo structure of the receptor.

**Questions:**

Please address the concerns in the weaknesses section.

Additionally, in Section 3/ 4.1 - How are the receptor and ligand graphs constructed - is it based on atom connectivity or distances between atoms?

---

> ### Author Response · Authors · 2023-11-17
> **Response to Reviewer ssPe (Part 1/2)**
>
> *We sincerely thank you for your time and efforts in reviewing our paper, and your valuable feekback. We are happy to see that you **recognize the novelty and effectiveness of prior-shifting and prior-conditioning mechanism** proposed in our paper. Please see below for our responses to your comments, and **the revised texts are denoted in red**.*
>
> **Q1: The paper cites Sattoras et al - and says it's an SE(3) Equivariant neural network - whereas it is E(N) GNN**
>
> A1: Thank you for your feedback. The architecture design of networks is not our primary focus of our paper. The backbone of networks utilized in our method just following with previous work [1-3]. And we acknowledge that EGNN [1] we used is an E(3)-equivariant neural network and chirality is also an important property of protein molecules. However, the input of the EGNN is a protein-ligand complex. Thus, for example, given a protein pocket (which is not symmetric in general), **a pair of stereoisomers will not share the same representation. This also follows TargetDiff [2]**. We appreciate your suggestion and will consider it in our future work.
>
> **Q2: The authors haven't shown their IPNet is SE(3) equivariant**
>
> A2: The graph in our IPNet is fully-connected, as detailed in the **Appendix-D.1**.
> However, even using the KNN graph, IPNet still maintains SE(3) equivariance. The proofs are provided as below:
> Since atom types are always invariant to SE(3)-transformation during the generative process, we only need to consider the atom coordinates.
> Recall the **equation 13 in the manuscript**, and we denoted $x_{t,i}^{C,l}- x_{t,j}^{C,l}$ as $d^{C, l}_{t,ij}$ for simplification. The atom hidden embeddings $h$ and coordinates $x$ can be updated as follow:
>
> $$\begin{aligned}
> h_{t,i}^{C,l+1}&=h_{t,i}^{C,l}+\sum_{j\in \mathcal N_C(i)}f_{h}^{C,l}(d_{t,ij}^{C,l},h_{t,i}^{C,l},h_{t,j}^{C,l},e_{ij}^{C}) \\\\
> x_{t,i}^{C,l+1}&=x_{t,i}^{C,l}+\sum_{j\in \mathcal N_C(i)}(x_{t,i}^{C,l}-x_{t,j}^{C,l})f_{x}^{C,l}(d_{t,ij}^{C,l},h_{t,i}^{C,l+1},h_{t,j}^{C,l+1},e_{ij}^{C})\cdot\mathbb 1_{mol}
> \end{aligned}$$
>
> First, it is easy to see $d_{ij}$ does not change with the 3D roto-translation $T_g$:
> $$\begin{aligned}
> \hat{d_{ij}^2} &= \left\Vert T_g(x_i) - T_g(x_j) \right\Vert^2 = \left\Vert (\mathbf R x_i + \mathbf b) - (\mathbf R x_j + \mathbf b) \right\Vert^2 = \left\Vert \mathbf R x_i - \mathbf R x_j \right\Vert^2 \\\\
> &=(x_i - x_j)^T \mathbf R^T \mathbf R(x_i - x_j) =(x_i - x_j)^T\mathbf I(x_i - x_j) = \left\Vert x_i - x_j \right\Vert^2 = d^2_{ij}
> \end{aligned}$$
> where $T_g$ is the group of SE(3)-transformation, which can be written explicitly as $T_g(x) = \mathbf R x + \mathbf b$, $\mathbf R \in \mathbb R^{3\times3}$ is the rotation matrix and $\mathbf b \in \mathbb R^3$ is the translation vector.
> Since $h_i$, $h_j$, $e_{ij}$ are initially obtained from atom and edge features, which are invariant to SE(3) transformation, we have $h_i^l$ is SE(3)-invariant for any $l=1,...,L$.
>
> Then, we can prove that $x$ updated from **equation 13** is SE(3)-equivariant as follows:
> $$\begin{aligned}
> \phi_{\theta 1}(T(x_t^{C,l})) &= T(x_{t,i}^{C,l}) + \sum_{j \in \mathcal N_C(i)}\left(T(x_{t,i}^{C,l})-T(x_{t,j}^{C,l})\right)f_{x}^{C,l}\left(d_{t,ij}^{C,l},h_{t,i}^{C,l+1},h_{t,j}^{C,l+1},e_{ij}^{C}\right)\cdot \mathbb 1_{mol} \\\\
> &=\mathbf R x_{t,i}^l + \mathbf b + \sum_{j \in \mathcal N_C(i)}\mathbf R(x_{t,i}^{C,l}-x_{t,j}^{C,l})f_{x}^{C,l}\left(d_{t,ij}^{C,l},h_{t,i}^{C,l+1},h_{t,j}^{C,l+1},e_{ij}^{C}\right)\cdot \mathbb 1_{mol} \\\\
> &=\mathbf R (x_{t,i}^l + \sum_{j \in \mathcal N_C(i)}\mathbf R(x_{t,i}^{C,l}-x_{t,j}^{C,l})f_{x}^{C,l}\left(d_{t,ij}^{C,l},h_{t,i}^{C,l+1},h_{t,j}^{C,l+1},e_{ij}^{C}\right)\cdot \mathbb 1_{mol}) + \mathbf b \\\\
> &= \mathbf R(x_{t,i}^{C, l}) + \mathbf b \\\\
> &= T(\phi_{\theta 1}(x_t^{C, l}))
> \end{aligned}$$
> where $\phi_{\theta 1}$ is our IPNet.
>
> **References:**
>
> [1] Satorras, et, al., E(n) Equivariant Graph Neural Networks, ICML, 2021.
>
> [2] Guan et al., 3d Equivariant Diffusion for Target-aware Molecule Generation and Affinity Prediction, ICLR, 2023a.
>
> [3] Guan et al.,DecompDiff: Diffusion Models with Decomposed Priors for Structure-Based Drug Design, ICML, 2023.
>
> [4] Schneuing et al., Structure-based Drug Design with Equivariant Diffusion Models, arXiv:2210.13695, 2022.
>
> [5] Peng et al., Pocket2mol: Efficient Molecular Sampling based on 3d Protein Pockets, ICML, 2022.
>
> [6] Luo et al., A 3d Generative Model for Structure-based Drug Design, NeurIPS, 2021.
>
> [7] Ragoza et al., Generating 3D molecules Conditional on Receptor Binding Sites with Deep Generative Models, Chem Sci, doi: 10.1039/D1SC05976A.
>
> [8] Liu et al., Generating 3d Molecules for Target Protein Binding, ICML, 2022.

---

> ### Author Response · Authors · 2023-11-17
> **Response to Reviewer ssPe (Part 2/2)**
>
> **Q3: It is unclear how S is trained to ensure the molecule is not distorted to impossible molecules**
>
> A3: The original diffusion process inherently introduces noise and denoises to the position of each atom, which can be treated as a shifting process to position of each atom in the previous step. Current molecular diffusion models [1-3] designs reconstruction-based training objective to maximally ensure that the molecule is not distorted to impossible molecules. We follow this and further enhance this shifting process by incorporating protein-ligand interaction prior obtained from a pre-trained IPNet model. These interaction prior preserves informative geometric relationships between intra-molecules and inter-molecules, thus facilitating the 3D molecular generation process. Theoretical proofs are provided in the **revised Appendix-B.4**, which demonstrate the potential of our method to achieve a better likelihood compared with previous diffusion-based methods. Moreover, the $S_t$ in our method represents the cumulative shift as time step $t$, which is 0 at the beginning and end of the diffusion process by setting the coefficient $k_t$ of $S_t$ as $\sqrt{\bar{\alpha}_t} \cdot (1-\sqrt{\bar{\alpha}_t})$. It means that our method specifically focuses on affecting the positions of each atom in the intermediate steps of the diffusion process without changing the initial and target distributions.
>
> **Q4: Using the pocket structure for the diffusion process is counter intuitive. Moreover, without the presence of the ligand, we only have access to the apo structure of the receptor.**
>
> A4: Indeed, as you rightly point out, it is necessary and practical to consider apo/holo structures. However, to the best of our knowledge, there are no existing methods for SBDD take apo structures into consideration. The main reasons are that these data is difficult to obtain, and its distribution is very unique, making this task extremely challenging. Therefore, we just follow the task definition and experimental setup as existing computational methods for SBDD [2-8], only considering holo structures. We really appreciate your suggestions and will consider them for future work.
>
> **Q5: How are the receptor and ligand graphs constructed - is it based on atom connectivity or distances between atoms?**
>
> A5: Yes, in our paper, we follow previous methods like TargetDiff and DecompDiff, representing each atom as a node and encoding the inter-atom distances as edge attributes in the construction of Protein-Ligand graphs.
>
> **References:**
>
> [1] Satorras, et, al., E(n) Equivariant Graph Neural Networks, ICML, 2021.
>
> [2] Guan et al., 3d Equivariant Diffusion for Target-aware Molecule Generation and Affinity Prediction, ICLR, 2023a.
>
> [3] Guan et al.,DecompDiff: Diffusion Models with Decomposed Priors for Structure-Based Drug Design, ICML, 2023.
>
> [4] Schneuing et al., Structure-based Drug Design with Equivariant Diffusion Models, arXiv:2210.13695, 2022.
>
> [5] Peng et al., Pocket2mol: Efficient Molecular Sampling based on 3d Protein Pockets, ICML, 2022.
>
> [6] Luo et al., A 3d Generative Model for Structure-based Drug Design, NeurIPS, 2021.
>
> [7] Ragoza et al., Generating 3D molecules Conditional on Receptor Binding Sites with Deep Generative Models, Chem Sci, doi: 10.1039/D1SC05976A.
>
> [8] Liu et al., Generating 3d Molecules for Target Protein Binding, ICML, 2022.

---

> ### Author Response · Authors · 2023-11-20
> **Gentle Reminder**
>
> Dear Reviewer ssPe,
>
> We sincerely appreciate the time and effort you have dedicated to reviewing our paper. Your insightful questions and valuable feedback have been extremely beneficial.
>
> **In response, we have (1) clarified the task definition and experimental settings of the existing AI-driven SBDD, (2) explained the underlying mechanism of prior-shifting and provided the corresponding theoretical proofs to address your queries and concerns.**
>
> As the discussion period is approaching its conclusion in two days, we kindly request, if possible, that you review our rebuttal at your earliest convenience.
>
> Your further comments and insights would be invaluable to us as we strive to enhance our work.
>
> Thank you once again for your invaluable contribution to our research.
>
> Warm Regards,
>
> The Authors

---

> > ### Comment · Reviewer_ssPe · 2023-11-22
> > **Response to Authors**
> >
> > Dear Authors,
> >
> > Thank you for the rebuttal.  Having gone through all the other reviews as well as your responses I will currently stick to my score of 5 - reasoning below - but happy to improve to an accept if some more explanation is provided (don't need any new experimental results due to timelines).
> >
> > I am completely satisfied with answers to questions 1 and 5 and partially to question 3 (however, in my opinion, reconstruction as just an objective doesn't entirely guard against impossible molecules nor does increased likelihood) please let me know if I am missing something here.
> >
> > With regard to Q4, i am not satisfied to the answer as - apo structure for receptor alone can be obtained from potentially PDB files of receptor alone (when available) or from ESM type model predictions (they have shown to generate apo structures in the absence of ligands).
> >
> > While this is not crucial for other models as they don't take the pocket structure as an explicit input - the proposed IPDiff does and hence can unfairly benefit the method.

---

> ### Author Response · Authors · 2023-11-22
> **Response to Reviewer ssPe**
>
> Dear Reviewer ssPe,
>
> Thanks for your further comments, we carefully response to your comments as follows.
>
> **Q1: I am partially satisfied with answers to question 3 (however, in my opinion, reconstruction as just an objective doesn't entirely guard against impossible molecules nor does increased likelihood) please let me know if I am missing something here.)**
>
> A1: The general purpose of existing generative models (including our models and other generative models [1-3]) is to maximally fit the distribution in the hypothetical space of real molecules. **All these generative models can not guarantee** all the generated molecules are valid unless we manually add rule-based constraints into the generative process.
> Moreover, the **primary goal in SBDD is to generate molecules that bind tightly** to the given pocket, and our method is mainly designed to improve existing generative models from this perspective.  And, if necessary, post-processing procedures can be conducted to filter out impossible molecules generated by our IPDiff or other generative models. As demonstrated in our proofs (please refer to the A6 in the response to Reviewer AJPX), our IPDiff exhibits **a higher likelihood than previous diffusion-based** SBDD methods, indicating that the molecules sampled from our model are more likely to be real and to bind tightly to the given pockets.
>
> **Q2: With regard to Q4, I am not satisfied to the answer as - apo structure for receptor alone can be obtained from potentially PDB files of receptor alone (when available) or from ESM type model predictions (they have shown to generate apo structures in the absence of ligands). While this is not crucial for other models as they don't take the pocket structure as an explicit input - the proposed IPDiff does and hence can unfairly benefit the method.**
>
> A2: Firstly, our IPDiff introduces a new pretraining-diffusion paradigm for SBDD. In the first stage, IPNet is pretrained to model general protein-ligand interaction priors, while the diffusion model in the second stage fully utilizes the priors provided by IPNet through two novel mechanisms *prior-shifting* and *prior-conditioning*. If the apo structure is available or the apo structures predicted by ESM type models are reliable, we can **easily extend our IPDiff (including IPNet and the new diffusion model) to these apo structures** by recursively or adaptively conduct our prior-shifting and prior-conditioning. Thanks for your constructive suggestions, we will further explore more complex scenarios for future work with our new pretraining-diffusion paradigm.
>
> Secondly, it's important to note that previous methods for SBDD, such as Pocket2Mol [1], DiffSBDD [2] TargetDiff [3], and DecompDiff [4], **all incorporate the protein pocket structures as an explicit input** during the molecule sampling process. Additionally, DecompDiff utilizes extra open-source software AlphaSpace [8] to obtain molecule templates for the protein-specific molecule generation process. We just strictly follow the task definitions and experimental settings of previous methods [1-7] to **ensure fair comparisons.**
>
>
>
> **References**
>
> [1] Peng et al., Pocket2mol: Efficient Molecular Sampling based on 3d Protein Pockets, ICML, 2022.
>
> [2] Schneuing et al., Structure-based Drug Design with Equivariant Diffusion Models, arXiv:2210.13695, 2022.
>
> [3] Guan et al., 3d Equivariant Diffusion for Target-aware Molecule Generation and Affinity Prediction, ICLR, 2023.
>
> [4] Guan et al.,DecompDiff: Diffusion Models with Decomposed Priors for Structure-Based Drug Design, ICML, 2023.
>
> [5] Luo et al., A 3d Generative Model for Structure-based Drug Design, NeurIPS, 2021.
>
> [6] Ragoza et al., Generating 3D molecules Conditional on Receptor Binding Sites with Deep Generative Models, Chem Sci, doi: 10.1039/D1SC05976A.
>
> [7] Liu et al., Generating 3d Molecules for Target Protein Binding, ICML, 2022.
>
> [8] Rooklin D et al., Alphaspace: fragment-centric topographical mapping to target protein–protein interaction interfaces, JCIM, 2015.
>
> ---
>
> Should there be any further points that require clarification or improvement, please know that we are fully committed to addressing them promptly.
>
> Thank you once again for your invaluable contribution to our research.
>
> Warm Regards,
>
> The Authors

---

> > ### Comment · Reviewer_ssPe · 2023-11-22
> > **Response**
> >
> > Dear Authors,
> >
> > Thank you again for the clarifications and the responses. Updated my score to a 6. Good luck!

---

> > > ### Author Response · Authors · 2023-11-22
> > > **Thanks for Your Support**
> > >
> > > Dear Reviewer ssPe,
> > >
> > > Many thanks for raising score! Your further comments and insights have been invaluable to our research.
> > >
> > > Warm Regards,
> > >
> > > The Authors

---

### Author Response · Authors · 2023-11-17
**Global Response**

We thank all the reviewers for the thorough reviews and valuable feedback. We are glad to hear that the idea is well-motivated and the method is novel (Reviewer ssPe, AJPX and F898), the paper is well-written and easy to follow (Reviewer qHa3 and F898), the experiments are sufficient and performance improvement shown in experiments are promising (all reviewers).
We have revised the manuscript according to the suggestions of all reviewers, and **marked revisions in red**.

We here summarize and highlight our responses to the reviewers:

1. We provide the theoretical derivation from the perspective of maximizing likelihood (Reviewer ssPe and AJPX) to explain the superiority of our IPDiff compared with previous diffusion-based SBDD methods.

2. We clarify the definition of SBDD tasks (Reviewer ssPe) and expermental settings (Reviwer qHa3 and AJPX) which are all followed with previous methods.

3. We add some experiments to demonstrate the effectiveness and soundness of our IPDiff (Reviewer qHa3, AJPX and F898). We also add more discussions about further application of IPDiff like its scalability (Reviewer AJPX) and extend it to other areas in the boarder AI comminity (Reviewer F898).

We response to each reviewer's questions below their reviews. Thank you and please feel free to ask any further questions.

---

### Comment · Area_Chair_tvaK · 2023-11-21
**Reviewers: Please respond to authors or update review**

Dear Reviewers,

The discussion phase will end tomorrow.  Could you kindly respond to the authors rebuttal letting them know if they have addressed your concerns  and update your review as appropriate? Thank you.

-AC

---

### Meta-Review · Area_Chair_tvaK · 2023-12-09

**Metareview:**

This paper proposes a diffusion method,Interaction Prior-guided Diffusion (IPDiff), for generating 3D molecules which bind to specific protein targets.   By using a pretrained protein-ligand interaction network IPNet in the forward noising process (prior shifting) and backward denoising process (prior conditioning), the authors ensure that molecular interaction information is incorporated into both the noising and denoising processes. The method is evaluated on CrossDocked2020.

Reviewers considered the method to be novel and well motivated with a clear theoretical derivation.  The method addresses an important problem and leverages PDBind pre-training to improve performance relative to previous approaches.   The empirical evaluation on the CrossDocked2020 benchmark shows SOTA results.  During discussion the authors clarified their method’s scalability, added evaluation on another benchmark and added and clarified metrics. While some reviewers maintain concerns about scalability due to the two-part training process and about the quality of the metrics this paper and other baselines use, this work is sufficient to demonstrate a valuable contribution which meets the bar in the field.

**Justification For Why Not Higher Score:**

- concerns about metrics and scalability

**Justification For Why Not Lower Score:**

- SOTA performance on benchmark
- novel and well motivated method

---

### Decision · Program_Chairs · 2024-01-16

Accept (poster)